# Death Induced by Survival gene Elimination (DISE) correlates with neurotoxicity in Alzheimer's disease and aging

Bidur Paudel[1], Si-Yeon Jeong[1,10], Carolina Pena Martinez[2], Alexis Rickman[2], Ashley Haluck-Kangas[1], Elizabeth T. Bartom [3,4], Kristina Fredriksen[5], Amira Affaneh[5], John A. Kessler [5], Joseph R. Mazzulli [5], Andrea E. Murmann[1], Emily Rogalski[6,7,11], Changiz Geula[6,7], Adriana Ferreira[8], Bradlee L. Heckmann[2], Douglas R. Green [9], Katherine R. Sadleir[5], Robert Vassar[5,6] & Marcus E. Peter [1,3] ✉

Alzheimer's disease (AD) is characterized by progressive neurodegeneration, but the specific events that cause cell death remain poorly understood. Death Induced by Survival gene Elimination (DISE) is a cell death mechanism mediated by short (s) RNAs acting through the RNA-induced silencing complex (RISC). DISE is thus a form of RNA interference, in which G-rich 6mer seed sequences in the sRNAs (position 2-7) target hundreds of C-rich 6mer seed matches in genes essential for cell survival, resulting in the activation of cell death pathways. Here, using Argonaute precipitation and RNAseq (Ago-RP-Seq), we analyze RISC-bound sRNAs to quantify 6mer seed toxicity in several model systems. In mouse AD models and aging brain, in induced pluripotent stem cell-derived neurons from AD patients, and in cells exposed to Aβ42 oligomers, RISC-bound sRNAs show a shift to more toxic 6mer seeds compared to controls. In contrast, in brains of "SuperAgers", humans over age 80 who have superior memory performance, RISC-bound sRNAs are shifted to more nontoxic 6mer seeds. Cells depleted of nontoxic sRNAs are sensitized to Aβ42-induced cell death, and reintroducing nontoxic RNAs is protective. Altogether, the correlation between DISE and Aβ42 toxicity suggests that increasing the levels of nontoxic miRNAs in the brain or blocking the activity of toxic RISC-bound sRNAs could ameliorate neurodegeneration.

Alzheimer's disease (AD) is characterized by neurodegeneration, but the specific events that cause neuronal dysfunction and death remain poorly understood. Plaques, composed of Aβ, and accumulation of hyper-phosphorylated tau protein (p-tau) into filaments and eventually neurofibrillary tangles (NFT) are at the base of the pathology of Alzheimer's disease (AD)[1]. Neuronal loss characterizes the end stage AD pathology and multiple cell death pathways have been implicated in this process including apoptosis, oxidative stress and mitochondria mediated cell death, necrosis, necroptosis, pyroptosis, ferroptosis, parthanatos, and autophagy[2–9]. Toxicity by amyloid precursor protein degradation fragments such as Aβ42 is well established[10] and a strong genetic association exists between early-onset familial forms of AD (FAD) and Aβ42[11]. More recently, it has been recognized that DNA damage accelerated by aging causes accumulation of somatic DNA alterations in individuals with AD[12,13]. An RNA component has not been described to contribute to the etiology of AD.

RNA interference (RNAi) is a form of post-transcriptional regulation exerted by 19-25 nt long double-stranded (ds) RNAs that negatively regulate gene expression at the mRNA level. The physiological way to induce RNAi is mostly mediated via miRNAs. They silence with the help of their seed at positions 2-7/8 of the miRNA guide strand[14]. The minimum seed length is 6 nucleotides[14,15]. Matching complementary regions (seed matches) predominantly located in the 3' untranslated region (3'UTR) of mRNAs are targeted[15] resulting in cleavage-independent translational silencing[16]. miRNAs are transcribed in the nucleus as primary miRNA precursors (pri-miRNA)[17] which are first processed by the Drosha/DGCR8 microprocessor complex into pre-miRNAs[18], and then exported from the nucleus to the cytoplasm[19]. Once in the cytoplasm, Dicer/TRBP processes them further[20,21] and these mature dsRNA duplexes are then loaded onto Argonaute (Ago) proteins to form the RNA-induced silencing complex (RISC)[22]. Consequently, the deletion of either Drosha or Dicer results in the loss of almost all miRNAs[23].

We previously discovered that any RISC-bound short RNA (R-sRNA) that carries a G-rich 6mer seed can kill cells by targeting essential survival genes in a RISC- and Ago2-dependent fashion ([24,25] and 6merdb.org). Cells die of simultaneous activation of multiple cell death pathways[26]. DISE (for Death Induced by Survival gene Elimination)[27] is a powerful evolutionary-conserved cell death mechanism active in humans and rodents. We found that the ratio of miRNAs with toxic versus nontoxic 6mer seeds determines their sensitivity to DISE[28]. The most abundant miRNAs are devoid of G-rich 6mer seeds and hence not toxic to cells[24]. However, certain tumor suppressive miRNAs (e.g. miR-34a/c-5p or miR-15/16-5p) use this mechanism to kill cancer cells[24,29]. We wondered whether there are diseases in which this anticancer mechanism is overly active, resulting in a reduced incidence of cancer but with tissue loss due to excessive cell death. We focused on AD as a candidate because advanced AD patients have lower cancer rates and late-stage cancer patients are less likely to develop dementia[30–32].

We now demonstrate a correlation between DISE and the DNA damage and neuronal cell death in AD and aging. We find that neurons from two mouse models of AD and induced pluripotent stem cell (iPSC)-derived neurons from AD patients exhibit a reduced median 6mer seed viability of R-sRNA. The aging mouse brain loses expression of nontoxic miRNAs. In addition, R-sRNAs from brains of a group of old individuals (>80 years) with memory capacity equivalent to those of 50-60 year-olds, often referred to as "SuperAgers," have a higher 6mer seed viability than R-sRNAs in the brains of control individuals. Our data demonstrate that exposure of differentiated SH-SY5Y (SH) cells to Aβ42 oligomers results in a shift of RISC-bound sRNAs (R-sRNAs) to more toxic 6mer seeds, and inhibiting RISC function or deletion of Ago2 reduces the toxicity of Aβ42 and blocks Aβ42-induced DNA damage. In addition, cells depleted of nontoxic sRNAs (e.g., lacking Drosha expression) are hypersensitive to Aβ42-induced death, and these cells can be protected against this toxicity by reintroducing miRNAs with nontoxic 6mer seeds. While our molecular manipulations of DISE likely have other effects, the observations as a whole suggest that increasing the levels of nontoxic miRNAs in the brain could lead to an unexplored approach to treating neurodegeneration.

## Results

### Association of reduced 6mer seed viability of R-sRNAs in the brains of AD mouse models with neurotoxicity

Throughout the manuscript we will be using two terms: 6mer seed viability and DISE. 6mer seed viability refers to the viability of three human and three murine cell lines after transfection with one of 4096 6mer seeds embedded in a neutral siRNA-like backbone that carries a 2'-O-methylation at positions 1 and 2 of the passenger strand to block RISC uptake[24,25]. The 6mer seed viabilities used are either based on the average % viability of the three human cell lines (for the analysis of

human cells/tissue) or the average % viability of the three murine cell lines (for the analysis of mouse cells/tissue), respectively. DISE refers to the mechanism of how the toxic 6mer seed containing sRNAs kill cells. sRNAs that carry a 6mer seed with low viability are more likely to kill cells through DISE while sRNAs that carry a nontoxic 6mer seed are potentially protecting cells from it by blocking access of DISE inducing sRNAs to the RISC.

The DISE mechanism is conserved between human and mouse[24,33]. To investigate whether DISE is associated with neurodegeneration seen in AD, we used two established mouse models replicating the primary hallmarks of AD pathogenesis, amyloid deposition and tau pathology. The first model is based on the widely used 5XFAD mice, which develop cerebral amyloid plaques and gliosis at two months of age, achieve high Aβ42 burden, and have reduced synaptic markers and a late neuron loss. However, 5XFAD mice do not form NFTs, do not show much of an increase in staining for p-tau[34,35], and, most relevant to this study, they do not show a significant degree of neuronal cell death up to 8 months of age[34]. Rubicon is a gene that constrains canonical autophagy[36] which when deleted in the brain results in increased levels of toxic Aβ oligomers[37]. 5XFAD Rubicon k.o. mice develop accelerated disease pathology and neurodegeneration, reactive microgliosis, tau pathology, and behavioral impairment similar to what is seen in human AD patients. The most significant difference between 5XFAD and 5XFAD Rubicon k.o. mice, however, is that they develop neuronal cell death early in the disease progression[38] (Fig. 1a, b).

We recently reported that a minor shift in the loading of toxic sRNAs into the RISC can cause activation of DISE, and that the balance of R-sRNAs with toxic versus nontoxic 6mer seeds (mostly miRNAs) correlates with cell fate[28] (Fig. 1c). To assess the role of DISE in disease situations we recently developed SPOROS, a semi-automated bioinformatics pipeline to analyze 6mer seed viability of sRNAs in a gene agnostic fashion[39]. The SPOROS bioinformatics pipeline which allows to visualize changes in total and R-sRNAs produces a number of graphical outputs[39]. Among them are: 1) A 6mer seed viability graph that plots sRNAs according to their abundance (Y axis) and 6mer seed viability (X axis), based on the average 6mer seed viability in three human or three mouse cell lines (see 6merdb.org) and 2) The median 6mer seed viability of all sRNAs in form of a box plot.

R-sRNA were identified by an Ago1-4 peptide pulldown, and RNA precipitation combined with high-throughput sequencing (Ago-RP-Seq) as described[28]. Performing a SPOROS analysis we compared the content of the RISC in mouse brains (cortex and hippocampus) of 6-month-old 5XFAD Rubicon$^{-/-}$ mice with advanced disease to the brains of three symptom-free age-matched 5XFAD Rubicon$^{+/-}$ mice. A number of miRNAs which constituted >96% of the R-sRNAs were significantly differentially expressed, most notably miR-9-5p (Fig. 1d). Interestingly, the median 6mer seed viability in the 5XFAD Rubicon$^{-/-}$ mice was significantly reduced (insert in Fig. 1d). This correlates with the increased Aβ42 levels in this model[38] and the resulting occurrence of cell death (Fig. 1a, b).

The second model we used represents a tauopathy. It has been shown that Aβ toxicity is dependent on the presence of tau[40,41] and tau depletion ameliorated Aβ-induced behavioral deficits when compared to transgenic mice overexpressing FAD-mutant amyloid precursor protein[40]. A growing body of evidence suggests that Aβ induces a series of post-translational tau modifications including its cleavage into a neurotoxic tau$_{45-230}$ fragment. High levels of this fragment were detected in human brains obtained from AD and other tauopathy subjects[42]. Human and murine tau are quite different and murine tau does not form NFTs[43]. However, transgenic mice expressing human tau$_{45-230}$ under the control of the Thy 1.2 promoter showed substantial cell death of pyramidal hippocampal neurons as early as 3 months after birth, progressive synaptic loss in the hippocampus, and behavioral deficits when compared to wild type controls[44]. Similar to the amyloid

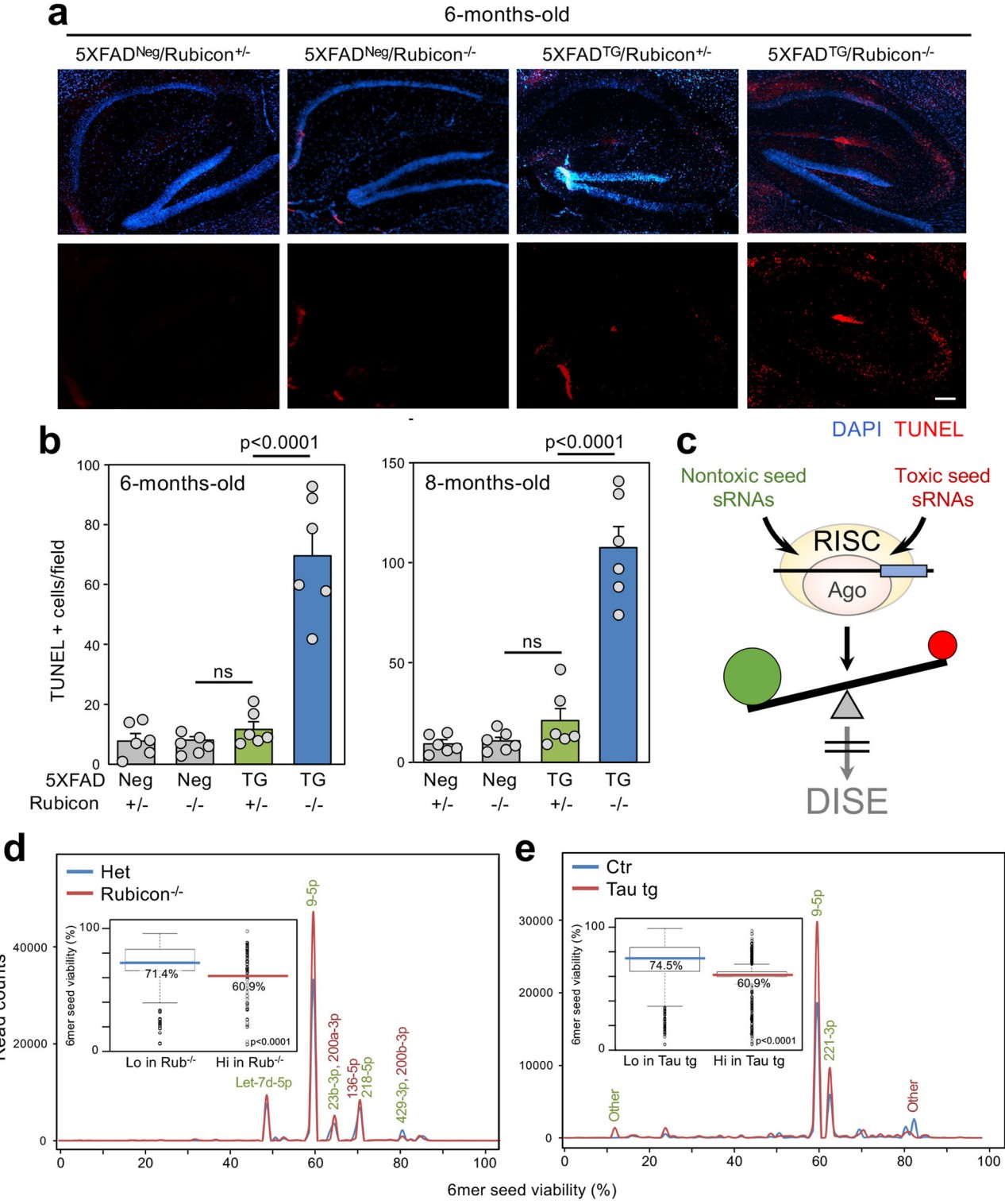

**a** 6-months-old

5XFAD$^{Neg}$/Rubicon$^{+/-}$ | 5XFAD$^{Neg}$/Rubicon$^{-/-}$ | 5XFAD$^{TG}$/Rubicon$^{+/-}$ | 5XFAD$^{TG}$/Rubicon$^{-/-}$

DAPI  TUNEL

**b** 6-months-old · 8-months-old

**c** Nontoxic seed sRNAs · Toxic seed sRNAs · RISC · Ago · DISE

**d** Het · Rubicon$^{-/-}$

**e** Ctr · Tau tg

field, it is now believed that smaller tau aggregates are more toxic than the bigger aggregates (i.e. NFT). The tau fragment expressed in the tau$_{45-230}$ tg mice was shown to aggregate into dimers and small oligomers that are highly toxic and present in AD patient brains[45]. Furthermore, it was shown that in the presence of this fragment full-length tau forms smaller and potentially more toxic aggregates. We therefore subjected brains (hippocampi) from 3 months tau$_{45-230}$ transgenic mice to the same analysis as the other mouse model (Fig. 1e). Consistent with the increase in cell death seen in this model, the median 6mer seed viability of R-sRNAs enriched or depleted in the brains of

these mice was reduced by more than 10% (insert in Fig. 1e). Like in the previous AD model miR-9-5p was again strongly enriched in the RISC in the brain of these mice compared to control mice corroborating previous findings on the role of miR-9-5p in neurodegeneration[46,47]. A comparison of the data from both mouse models suggested that the Ago-RP-Seq method reproducibly pulls down the same miRNAs from mouse brains. In fact, a majority of RISC-bound miRNAs was found to be deregulated in both mouse models (shared miRNAs labeled in black in Supplementary Fig. 1). The results obtained from the female 5XFAD Rubicon k.o. and the male tau$_{45-230}$ transgenic mice available to us

**Fig. 1 | Reduction in 6mer seed viability of R-sRNAs in AD mouse models with neurotoxicity. a** Representative TUNEL staining of brains of 6-month-old control (5XFAD^Neg/Rubicon^+/- and 5XFAD^Neg/Rubicon^-/-), 5XFAD tg (5XFAD^TG/Rubicon^+/-), and 5XFAD^TG/Rubicon^-/- mice. Brain sections were counterstained with DAPI. Size marker = 200 μm. **b** Quantification of TUNEL positivity in brains of mice with four different genotypes. Object counts were recorded in three different fields. Shown are the mean and SE of counts from six different mice for each genotype. Left, 6-months and right, 8-month-old mice. Student's two-sided t-test p-values and standard error values are given. ns, not significant. **c** Scheme illustrating how the ratio of RISC-bound sRNAs with nontoxic (green) versus toxic 6mer seeds (red) can protect cells from DISE. Ago, argonaute proteins. **d, e** 6mer seed viability graphs of differentially expressed R-sRNAs (p < 0.05) between 5XFAD Rubicon^+/- (Het) and 5XFAD Rubicon^-/- (**d**) and control and tau$_{45-230}$ transgenic mouse (**e**) brains. miRNAs are labeled that have a difference in expression of 1000 or more reads. Enriched sRNAs are shown in green and depleted sRNAs in red. Shown are averages of triplicate (**d**) or quadruplicate (**e**) samples. Inserts: Box and whisker plots of the 6mer seed viability of the same samples. Shown are the median viabilities of opposing differentially expressed and/or RISC-loaded sRNAs in the color of each sample/condition. Kruskal-Wallis median test p-value is given. Definition of box and whisker plots in d and e: The lower and upper hinges of the box correspond to the first and third quartiles (the 25th and 75th percentiles). The upper/lower whiskers extend from the upper/lower hinge to the largest/smallest value no further than 1.5 x inter-quartile range (distance between the first and third quartiles) from the hinge. Data beyond the end of the whiskers are outliers and are plotted individually.

were quite similar suggesting no major difference between male and female mice.

We do not believe that the observed difference between normal and AD brains is due to an increase in infiltrating immune cells because an analysis of neuron, glia and macrophage/lymphocyte specific miRNAs[48–51] bound to the RISC did not support that interpretation (Supplementary Fig. 2). In summary, the analysis of the R-sRNAs from the brains of two mouse models of dementia with signs of neuronal cell death suggests a possible association of DISE with the neurotoxicity seen in these models.

### Reduced 6mer seed viability of R-sRNAs in AD-derived iPSC neurons

The data on the AD mouse models suggested that a lower 6mer seed viability may be associated with an increase in neurotoxicity. To determine whether this could also be found in human neurons, we analyzed iPSC-derived neurons generated from two control and two AD patients (Fig. 2a, b). AD-derived iPSC neurons had highly upregulated Tau1 protein and showed signs of ongoing DNA damage stress with increased H2AX phosphorylation (Fig. 2c). In the AD-derived cells when assessing the total R-sRNA changes a number of miRNAs with nontoxic 6mer seeds were downregulated compared to control cells (Fig. 2d) resulting in a drop in median 6mer seed viability in the sRNAs that were enriched in the AD samples (Fig. 2e) suggesting that in cells from AD patients nontoxic R-sRNAs may be reduced.

### 6mer seed viability of R-sRNAs drops during aging

miRNAs have been shown to protect neurons from cell death[52,53], but aging neurons may be less able to generate miRNAs[54]. Such aging neurons could be more susceptible to small RNAi active RNAs that induce cell death. To test whether aging neurons would lose their protection by nontoxic miRNAs, we compared the RISC content of the brain cells of young (2-month-old) with that of old (18-month-old) mice (Fig. 3a). We found that the RISC in brain cells (cortex and hippocampus) of older mice contained significantly less sRNAs with high 6mer seed viability. This was independently confirmed in an analysis of another set of young and old mice (Supplementary Fig. 3a–e). The miRNAs that were pulled down in these two experiments were very similar (shared miRNAs labeled in black in Supplementary Fig. 3a, b), median 6mer seed viabilities and 6mer seed composition of all R-sRNAs in these samples were indistinguishable (Supplementary Fig. 3c, d), and a Pearson correlation demonstrated high reproducibility between the two experiments (Supplementary Fig. 3e). Consistently, in experiment #2, similar to the observation in experiment #1, the median 6mer seed viability of the differentially expressed R-sRNA was lower in the brains of old mice compared to young mice (Supplementary Fig. 3f).

How could aging neurons lose expression of mostly nontoxic miRNAs? A progressive loss of Dicer and the resultant global decrease in miRNA expression with age has been suggested as one mechanism. A study showed that Dicer expression in the brains of two year old mice is substantially reduced compared to the brains of 6-week-old mice[54].

We therefore wondered whether neurons with increasing age would lose expression of some of their nontoxic miRNAs and whether sRNAs that are enriched in older neurons could be more toxic, rendering the neurons primed for DISE. We used two approaches to address the question of whether cells with age would lose nontoxic miRNAs:

1. We tested whether the loss of nontoxic R-sRNAs can be seen in aged neurons in vitro. iPSC-derived neurons were aged in a dish as described[55] (Supplementary Fig. 4). These cells were cultured for up to 190 days in poly-d-lysine and laminin coated dishes. We detected a substantial loss of Dicer protein with increasing age (right insert in Fig. 3b). An Ago-RP-Seq-SPOROS experiment suggested a small but significant decrease in 6mer seed viability in the 5-month-old versus 1-month-old iPSC-derived neurons (left insert in Fig. 3b). While the difference seems small, we would like to point out that the analysis shown here is of the total R-sRNAs and not of the R-sRNAs that are differentially expressed.

2. We analyzed a data set of a study that had reported major changes in the miRNA content of peripheral blood mononuclear cells (PBMCs) between young and old cohorts[56] because it was shown before that besides the brain spleen/immune cells is another tissue whose miRNA content changes with age[54]. A reanalysis of these data for the 6mer seed viability of the differentially expressed mature miRNAs revealed a substantial drop in median 6mer seed viability and a concomitant increase in G content of the 6mer seeds in the miRNAs upregulated in the PBMCs from the old cohort (Fig. 3c). This is in line with our data on the mouse brains and suggests that with age certain tissues lose nontoxic miRNAs and may be primed to undergo DISE.

If the assumption is correct that a higher 6mer seed viability of R-sRNAs in brain cells protects from neuronal death and dementia and this is reduced with increasing age, then individuals that are known to be less prone to AD or other forms of dementia may maintain a higher level of nontoxic miRNAs in the RISC similar to younger individuals. To test this hypothesis, we analyzed the RISC content of brain cells from SuperAgers which are individuals age 80 and above whose episodic memory performance is at least at the level of cognitively average individuals in their 50 s and 60 s[57]. We obtained postmortem brain tissue from three SuperAgers and compared them to three cognitive normal gender matched individuals of roughly the same age (Supplementary Table 1). We did indeed find that the RISC in the SuperAger brains contained sRNAs with a significantly higher 6mer seed viability than in control brains (Fig. 3d). Similar results were observed when we individually analyzed the data from the three brains in each group (Supplementary Fig. 5), suggesting that the minor difference in age between the two groups did not impact the results. These preliminary data suggest that SuperAgers may have a greater resilience to the damage by toxic sRNAs which may be associated with dementia such as AD. Our data show that the R-sRNA 6mer seed viability measurements correlate with both decreased (AD pathology) and increased (SuperAger) neuronal function.

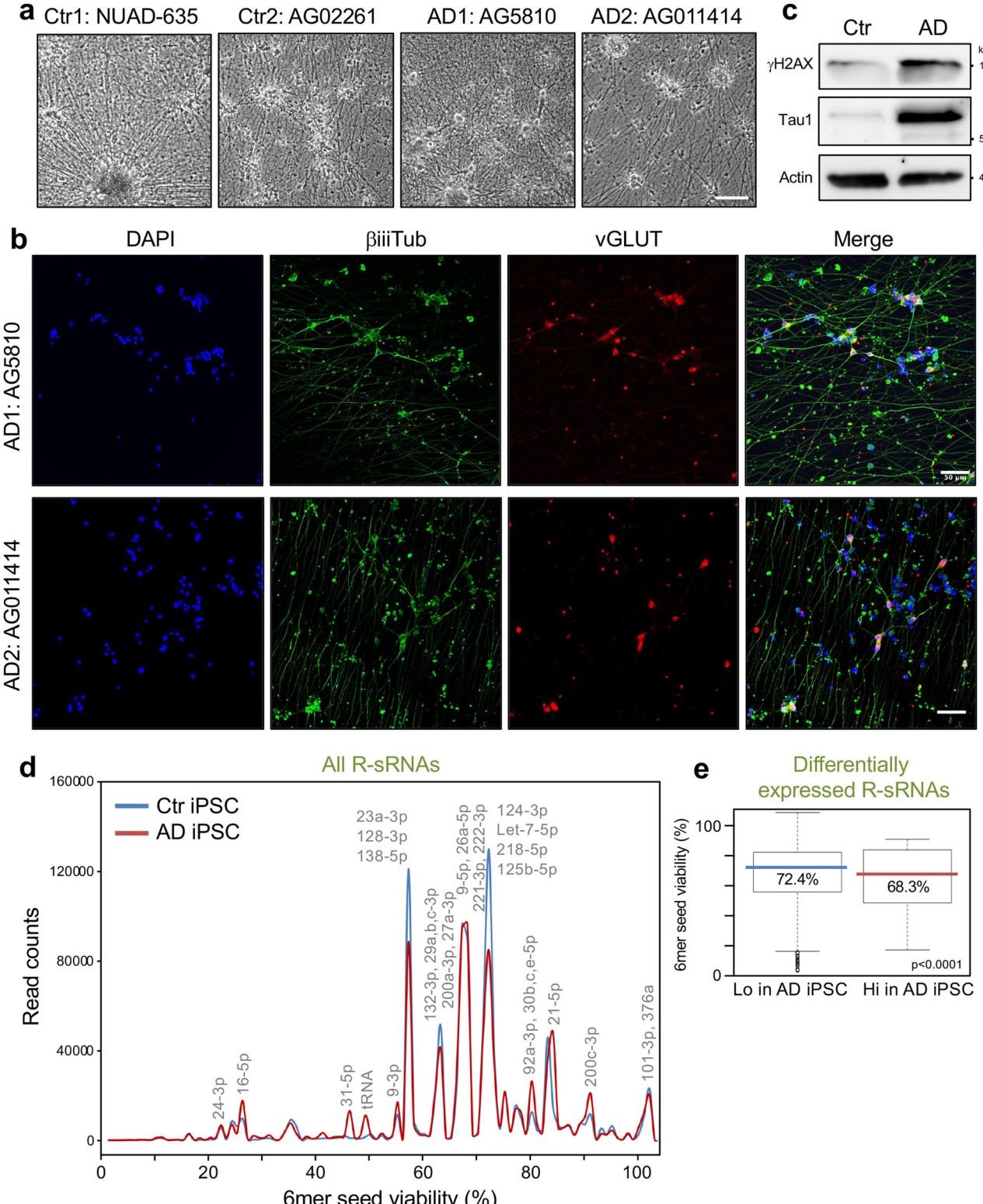

**Fig. 2 | Reduced 6mer seed viability of R-sRNAs in iPSC-derived neurons from AD patients. a** Bright field image of the NGN2 excitatory neurons derived from two control and two AD patients 36 days post differentiation. Size bar = 500 μm. **b** Immunofluorescence images of the AD-derived neurons using the neuronal marker βiiiTubulin and the glutamatergic neuronal marker vesicular glutamate transporter (vGLUT) counterstained with DAPI. Size bar = 50 μm. **c** Western blot analysis of control (Ctr) (NUAD0635) and AD (AG5810) iPSC-derived neurons. The analysis was done with half of the cells that were subjected to the Ago-RP-Seq-SPOROS analysis. **d** 6mer seed viability graph of total R-sRNAs of the two Ctr and two AD patient-derived iPSC-derived neurons. Shown are averages of duplicate samples. sRNAs with 10,000 or more reads in at least one sample are labeled. In each label sRNAs are listed in the order of expression levels. **e** Box and whisker plots of the differential 6mer seed viability of the samples in **d**. Medians (in the color of each sample/condition) and Kruskal-Wallis median test p-value are given. For definition of box and whisker plot see legend of Fig. 1.

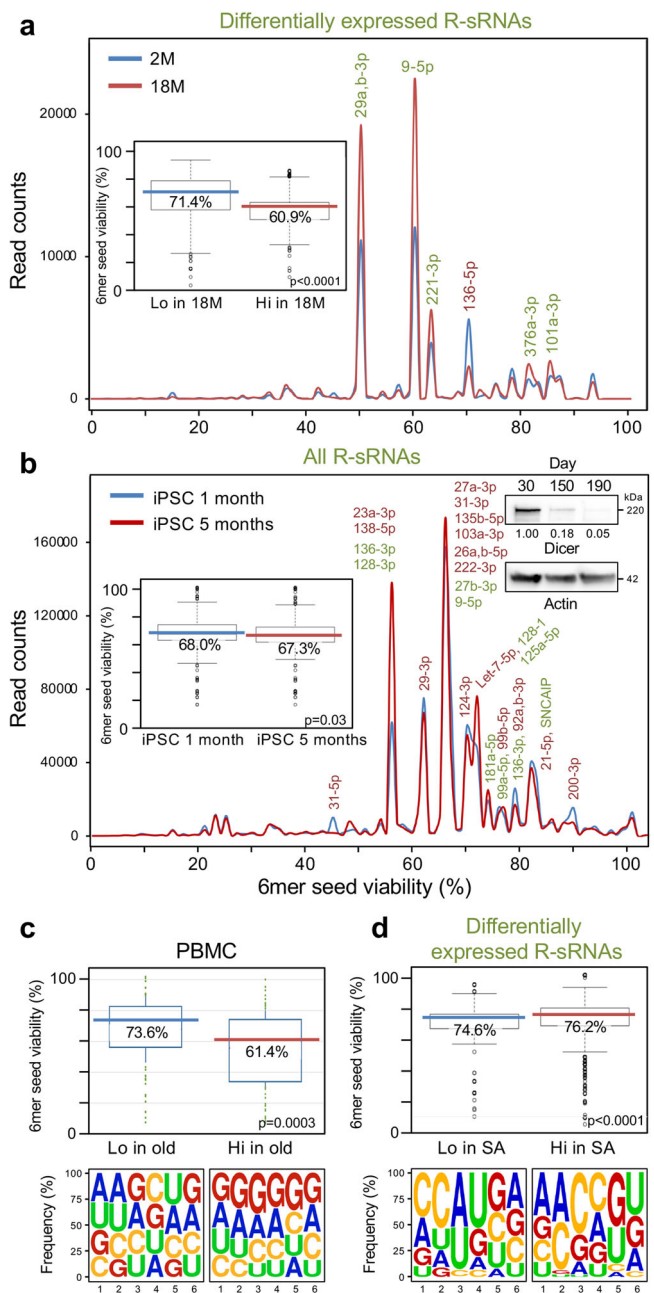

**Fig. 3 | Changes in 6mer seed viability of R-sRNAs with age. a** 6mer seed viability graph of differentially expressed R-sRNAs of 2- (2 M) and 18- (18 M) month-old mouse brains. Shown are averages of triplicate samples. sRNAs that have a difference in expression of 1000 or more reads are labeled. Enriched sRNAs are shown in green and depleted sRNAs in red. Insert: 6mer seed viability plot of the differential 6mer seed viability of the analysis. **b** 6mer seed viability graph of total R-sRNAs of iPSC-derived neurons aged for 1 and 5 months (one sample each). sRNAs with 10,000 or more reads are labeled. In each label sRNAs are listed in the order of expression levels. Enriched sRNAs are shown in green and depleted sRNAs in red. Insert: 6mer seed viability plot of the differential 6mer seed viability of the analysis. Western blot insert: Analysis of iPSC-derived neurons aged in vitro for 30, 150 or 190 days. Densitometrical quantification relative to the 30-day time point and actin is given. **c** Top, Average 6mer seed viability of all significantly up- and down-regulated miRNAs in PBMCs of 11 old versus 11 young donors[56]. Bottom, Nucleotide composition at each of the six seed positions in the miRNAs analyzed above. **d**, Top, 6mer seed viability plots of the differential 6mer seed viability of postmortem cognitive normal (CN) and SuperAger brains. Age of the three CN participants: 100, 87, 83 - average = 90, age of the three SuperAgers: 82, 90, 85 - average = 85.7. Bottom, average 6mer seed composition of the sRNAs in the same samples. Medians (in the color of each sample/condition) and Kruskal-Wallis median test p-value are given in the box plots in a-d. For definition of box and whisker plot see legend of Fig. 1.

the more physiologically relevant protocol of differentiating SH cells with RA and BDNF for all subsequent experiments.

To determine whether the toxicity of Aβ42 to SH cells was associated with a shift to fewer nontoxic RISC-bound reads, we treated 7-day differentiated SH cells with Aβ42 for 2, 4, and 6 hours. We chose these time points to capture changes before the initiation of cell death, which was first detectable 24 hrs after treatment (Fig. 4c). Employing the Ago-RP-Seq-SPOROS pipeline we found that the sRNAs that were significantly enriched in the RISC of cells treated with Aβ42 compared to cells treated with Aβ40 had a significantly lower 6mer seed viability than sRNAs that were depleted in the RISC (Fig. 4d).

## Modulating RISC content alters cells sensitivity to Aβ42-induced toxicity

We previously reported that global downregulation of most abundant miRNAs, which mainly carry nontoxic 6mer seeds, caused by deleting either Dicer or Drosha sensitizes cells to effects of toxic sRNAs[24,27,64–66]. To determine whether neuron-derived cells gain sensitivity to Aβ42-induced cell death after removal of most miRNAs, we generated Drosha k.o. clones of the neuroblastoma cell line NB7 (Fig. 5a). This resulted in about 50% reduction of RISC-bound miRNAs (Fig. 5b) with major miRNAs substantially reduced (Fig. 5c). Previously, we characterized Drosha k.o. HCT116 cells and reported that in the absence of nontoxic miRNA expression most R-sRNAs were not miRNAs. With Ago2 expression levels were unaffected in these cells, the median 6mer seed viability across all R-sRNAs was much lower than that of R-sRNAs in wt cells[65].

Thus, to investigate the effect of lack of Drosha on the brain-derived cells, we sequenced the total cellular sRNAs from the parental cells and two Drosha k.o. clones and analyzed the sequencing data with SPOROS. Similar to the data obtained with Drosha k.o. HCT116 cells, in the two NB7 k.o. clones, many canonical miRNAs were reduced in the total RNA and other sRNAs (i.e., rRNA and tRNA fragments) were quite prominent (Supplementary Fig. 7a). This resulted in a significant drop in median 6mer seed viability in both k.o. clones (Supplementary Fig. 7b) caused by the loss of miRNAs which mostly carry nontoxic 6mer seeds. Consistent with previous data, we found that the two k.o. clones were much more sensitive to DISE induced by sGGCAGU transfection (Supplementary Fig. 7c).

After establishing that NB7 cells with reduced Drosha expression were hypersensitive to DISE, we tested whether the two k.o. clones were also more susceptible to cell death induced by Aβ42 than the parental cells. Parental NB7 cells were moderately sensitive to Aβ42

## Toxic Aβ42-oligomers shift the balance of RISC-bound short RNAs to lower 6mer seed viability

To determine whether DISE could be involved in the death of brain-derived cells, we chose the well characterized SH-SY5Y (SH) cell line differentiated into neurons to model neurotoxicity upon treatment with Aβ42 oligomers[58–60]. Similar to other brain-derived cell lines[25], SH cells were susceptible to sRNAs that carry toxic 6mer seeds such as an sRNAs carrying the 6mer seed of miR-34a-5p (GGCAGU)[24] or the consensus toxic 6mer seed GGGGGC[25] (Supplementary Fig. 6a). While moderately sensitive, SH cells gain sensitivity to toxic Aβ42 oligomers after differentiation[61]. We demonstrated that our preparation of Aβ42 oligomers was soluble without aggregates (not shown) and was more active than control Aβ40 in SH cells fully differentiated for 7 days either by treating them with 1 mM dbcAMP[62] (Supplementary Fig. 6b, c) or by using a differentiation protocol involving retinoic acid (RA) and brain-derived neurotrophic factor (BDNF)[63] (Fig. 4a–c). After establishing the specific activity of Aβ42 oligomers, we decided to use

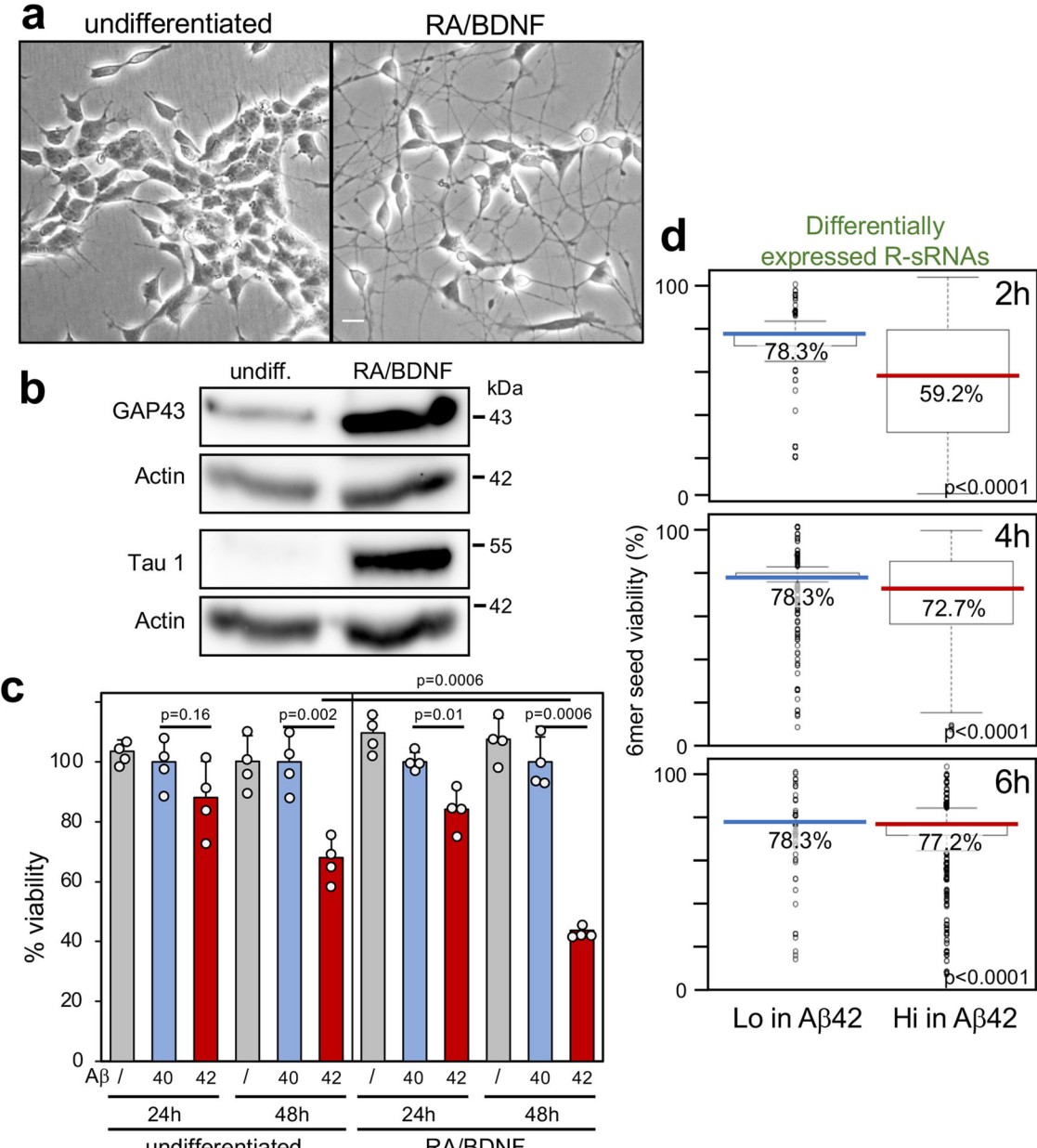

**Fig. 4 | Exposure of differentiated SH cells to Aβ42 results in a shift to R-sRNAs with lower 6mer seed viability. a** Phase contrast image of undifferentiated SH cells or SH cells differentiated with RA/BDNF for 7 days. Size bar = 60 μm. **b** Western blot analysis of cells in **a**. This is one of three independent experiments. **c** Viability of undifferentiated or 7 day differentiated SH cells treated with either Aβ40 or Aβ42 for 24 or 48 hrs. Data were normalized to the treatment with Aβ40. Mean and SD of quadruplicate samples are shown. Shown is one of four biological repeats. p-values represent two-sided Student's t test **d** Difference in 6mer seed viability of R-sRNAs significantly (p < 0.05) under- or overrepresented in 7 day differentiated SH cells treated with Aβ42 for different periods of times (compared to Aβ40 treated cells). Shown are the medians (in the color of each sample/condition) and Kruskal-Wallis median test p-values. This is one of two similar independent experiments. For definition of box and whisker plots see legend of Fig. 1.

relative to Aβ40 (Fig. 5d). In contrast, both Drosha k.o. clones were ~10% more sensitive to Aβ42-induced cell death consistent with the hypothesis that miRNAs with nontoxic 6mer seeds could protect cells from this toxicity.

Having provided evidence of an association of DISE with Aβ42-induced cell death we wondered whether we could protect NB7 Drosha k.o. cells by reconstituting them with some of the nontoxic miRNAs most downregulated in these cells. To identify such miRNAs, we performed a SPOROS analysis of R-sRNAs in wt and Drosha k.o. cells (Fig. 5e). While a few miRNAs with toxic 6mer seeds were increased in these cells (i.e., miR-15/16-5p), there were a larger number of miRNAs with nontoxic seeds that were strongly reduced in the two k.o. clones

resulting in a clear drop in 6mer seed viability and in a more C-rich average 6mer seed in the R-sRNA of the k.o. clones (Fig. 5f, g). The three most highly downregulated and nontoxic miRNAs miR-21-5p, miR-25-3p, and miR-92a-3p were also the top three most abundant ones in the RISC of wt cells (Fig. 5h). In fact, these three miRNAs comprised 21.2% of all RISC-bound miRNAs in the wt cells and only 7.5% and 9.7% in the two Drosha ko clones, respectively. We therefore decided to reconstitute the k.o. cells with these three miRNAs to test whether adding them back to the RISC would render cells more resistant to the toxicity exerted by Aβ42.

To test this, we transfected NB7 Drosha k.o. cells with the three miRNAs individually or with a cocktail of all three. Because any

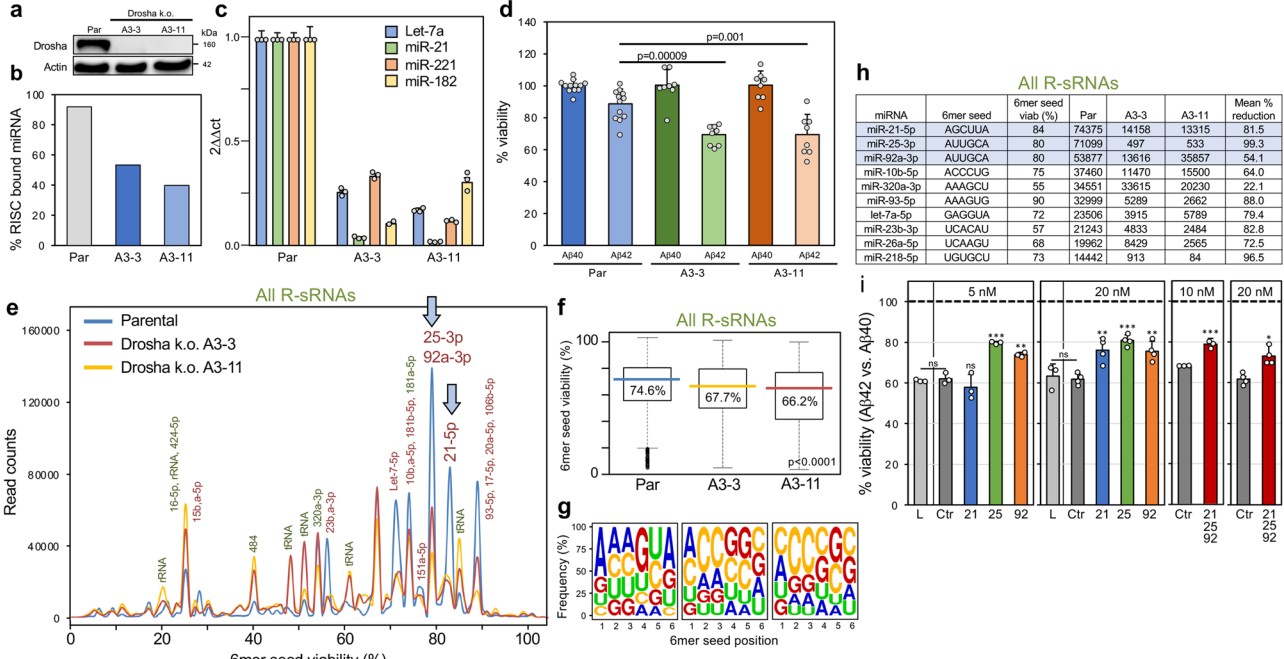

**Fig. 5 | Increasing the amount of miRNAs with nontoxic 6mer seeds lowers the sensitivity of cells to Aβ42-induced cell death. a** Western blot analysis of parental (Par) NB7 cells and two Drosha k.o. clones. **b** Relative amounts of total RISC-bound miRNAs of the cells in a. Totals counts of duplicate samples are shown. **c** Real-time PCR quantification of major miRNAs in the two k.o. clones relative to parental cells. Shown are averages with variance of samples done in duplicate (miR-221/miR-181) or SD of samples done in triplicate (Let-7a and miR-21). The data on miR-21 and Let-7a are representative of two biological replicates. **d** Viability assay of parental NB7 cells or two Drosha k.o. clones after a 48 hr treatment with Aβ40 or Aβ42. Shown are the Aβ40 normalized data points from for parental cells (n = 12), A3-3 (n = 8) and A3-11 (n = 8), a combination of two biological replicates. Mean with SD and two-sided Student's t-test p-values are shown. **e** 6mer seed viability graph of all R-sRNAs sequenced in parental NB7 cells and two Drosha k.o. clones. sRNAs with 10,000 or more reads in at least one sample are labeled. Enriched sRNAs are shown in green and depleted sRNAs in red. The peaks containing the most highly expressed and most downregulated miRNAs in the two k.o. clones are marked by a blue arrow.

**f** Box plots of the 6mer seed viability of the samples in **e**. Shown are the medians (in the color of each sample/condition) and Kruskal-Wallis median test p-value.
**g** Average 6mer seed composition of all sRNAs in the three samples in **e**. **h** The top 10 most abundant RISC-bound miRNAs in parental NB7 cells with their 6mer seed and 6mer seed viabilities (%) and their expression and expression changes in the two Drosha k.o. clones. The top three most abundant miRNAs (labeled in **e** with arrows) are highlighted in blue. **i** Viability assay of a 1:1 mix of the two Drosha k.o. clones cells treated for 72 hrs with 20 μM Aβ42 relative to cells treated with Aβ40 (stippled line). Cells were mock transfected (L) or transfected with a control sRNA modified to block RISC access, or different amounts either miR-21-5p (21), miR-25-3p (25), miR-92-3p (92) or a cocktail of the three miRNAs for 24 hrs. Total concentration for each miRNA or the cocktail is given. The cocktail contained equal amounts of the individual miRNAs. Mean with SD and two-sided Student's t-test p-values are shown (**d, i**). * p < 0.05, ** p < 0.01., *** p < 0.001. This is representative of three independent similar experiments.

nontoxic control sRNA could be incorporated into the RISC and block its activity, we compared this with cells treated with only lipid or transfected with a control sRNA modified on both strands at position 1 and 2 with 2'-O-methylation (sNT) which we previously showed to completely prevent RISC loading[64,66]. In this experiment we found that introducing the three miRNAs individually or as a cocktail rendered cells ~10% less sensitive to Aβ42 (Fig. 5i). The fact that all three miRNAs protected cells from Aβ42-induced cell death to a similar degree is consistent with a model that it is not the activity of each individual miRNA and the targeting of its targets alone that determines cell fate but rather the fact that they carry nontoxic 6mer seeds and are blocking RISC access.

**Inhibiting RISC ameliorates Aβ42-induced toxicity**
Our data suggest an association between neurotoxicity seen in AD and the 6mer seed viability of R-sRNAs. We wondered whether mechanistically, Aβ42, the primary mediator of neurotoxicity in AD induces cell death through RISC and RNAi. We therefore tested the involvement of Ago2 in the toxicity elicited by Aβ42. We first used the Ago2 inhibitor aurintricarboxylic acid (ATA)[67] to determine whether toxic RNAs that kill cells through Ago2 showed reduced activity in neuron-derived cells with inhibited RISC. We treated NB7 cells with a low concentration of ATA and showed that this inhibited DISE induced by either the two toxic 6mer seed-containing sRNAs used above or by the

highly toxic Huntington's disease (HD)-derived CAG trinucleotide repeat containing sRNA sCAG which we recently described[64,68] (Supplementary Fig. 8a). ATA inhibited toxicity induced by all three sRNAs in a dose-dependent manner. Undifferentiated SH cells (used because their growth could be monitored by an IncuCyte analysis) were protected from sCAG-induced cell death by ATA (Supplementary Fig. 8b) and a rescue of sCAG-induced cell death was also observed in differentiated cells (Supplementary Fig. 8c).

To determine whether ATA which at higher concentrations inhibits multiple endonucleases[69] did indeed act by inhibiting Ago2 and hence RNAi, we performed an Ago-RP-Seq analysis of NB7 cells transfected with either 1 nM sNT1, sGGGGGC or sGGGGGC in the presence of a low concentration of 10 μM ATA (Supplementary Fig. 8d). ATA treatment substantially prevented RISC uptake of sGGGGGC. This analysis confirmed the activity of ATA on Ago2 and suggested that the inhibition of the RISC uptake of the toxic sRNA is what protects the cells from transfection with sGGGGGC.

After establishing that ATA could protect cells from DISE induced by a toxic sRNA, we tested whether ATA which was previously shown to protect primary neurons from Aβ42-induced death[70] had an effect on Aβ42-induced cell death in differentiated SH cells at 10 μM (Fig. 6a). A significant 10% reduction in cell death was observed. To exclude that the effects seen with ATA were due to inhibition of nucleases other than Ago2, we deleted Ago2 in SH cells. Three clones with very low or

undetectable Ago2 expression were isolated (Fig. 6b). Ago2 deficient cells were substantially resistant to the toxicity of transfected sCAG (Supplementary Fig. 9). Knock-out of Ago2 did not affect the ability of the cells to differentiate as shown by the induction of the synaptic marker GAP43 (Fig. 6c). However, we used undifferentiated cells for the following experiments involving these k.o. cells to prevent any effect of reduced differentiation due to impaired RISC/miRNA activity. Similar to the effects observed with ATA treatment all three Ago2 k.o. clones were less sensitive to Aβ42-induced toxicity compared to parental cells (Fig. 6d). Neither ATA treatment nor deletion of Ago2 completely blocked Aβ42-induced toxicity. This could be due to the activation of RISC-independent pathways or an involvement of the other three Ago proteins expressed in human cells mediating toxicity of sRNAs induced by Aβ42.

## Aβ42 oligomer-induced DNA damage is dependent on tau and an active RISC

Enoxacin is an antibiotic that can cause an upregulation of miRNAs by stabilizing Dicer/TRBP[71,72] and treatment with Enoxacin has been shown to improve survival of cultured dopaminergic neurons[73]. We therefore wondered whether Enoxacin could protect cells from the toxicity of Aβ42. To test this, we first pretreated differentiated SH cells with Enoxacin for 24 hours and exposed the cells to Aβ40/42 for another 24 hrs (Fig. 6e), a time frame during which substantial DNA damage occurred as seen by an increase in phosphorylation of H2AX (γH2AX) (Fig. 6f). Surprisingly, Enoxacin had little effect on cell viability (Fig. 6e, bottom) however, it seemed to completely block DNA damage induced by Aβ42 at a concentration of 20 μM or higher (Fig. 6e, top). Enoxacin has also been reported to enhance DNA repair after DNA damage through boosting Dicer activity[74]. This activity was shown to involve the production of Dicer-dependent non-coding RNAs called DNA damage response RNAs (DDRNAs)[75]. To test whether this alternative activity of Enoxacin on Dicer contributed to the reduction of DNA damage after Aβ42 exposure, we first treated differentiated SH cells for 24 hours with Aβ42 and then added Enoxacin for another 2 hours. Regardless of whether we left Aβ42 on the cells (Fig. 6g, left) or washed it out (Fig. 6g, right), a short incubation with Enoxacin substantially reduced γH2AX levels suggesting that the reduction of DNA damage is at least in part caused by the activity of Enoxacin to accelerate Dicer mediated DNA repair. Overall, the data suggested that DNA damage is not the cause of the cell death induced by Aβ42 but rather a result of it. The separation of DNA damage and Aβ42-induced toxicity now allowed us to mechanistically interrogate the pathway of Aβ42-induced cell death and the role of the RNAi and DNA damage in it.

Tau has been shown to be essential for toxicity of Aβ42 in various models[40,41]. To investigate whether tau was also required for Aβ42-induced cell death observed in SH cells, we tested tau k.o. SH cells[76]. Interestingly, tau k.o. cells still differentiated normally (Fig. 6h). The two differentiated k.o. clones showed reduced sensitivity to Aβ42-induced toxicity and clone 231 K (used in Fig. 6h) was almost completely resistant (Fig. 6i). As expected from the above data, Aβ42-induced DNA damage was blocked in the tau k.o. cells (Fig. 6j). This analysis placed tau downstream of Aβ42 and upstream of cell death and DNA damage. To determine where DNA damage occurs with respect to the RISC activity, we pretreated Aβ42 exposed differentiated SH cells with 10 μM ATA. This ATA concentration which only reduced cell death by ~10% (see Fig. 6a) completely blocked DNA damage (Fig. 6k) and a similar finding was made when the Ago2 k.o. clones were tested (Fig. 6j). This places DNA damage downstream of the RISC and is consistent with our report showing that DISE inducing RNAi active sRNAs cause DNA damage in brain-derived cells detectable in form of H2AX phosphorylation[26]. In summary, together our data suggest that DISE correlates with the pathology of AD initiated by the main mediator of AD, toxic Aβ oligomers. With an involvement of tau the RISC is engaged, enhancing the toxic activity of Aβ42 through

shifting the balance of RISC-bound sRNAs to more toxic 6mer seeds. This association links DISE with DNA damage and in conjunction with RISC-independent pathways this may result in neuronal dysfunction and neurodegeneration.

## Discussion

DISE was discovered as a powerful cell death mechanism[33]. Our analysis of R-sRNAs in various AD models now supports DISE as a mechanism that may affect the survival of neurons. While our analysis included any sRNA that can enter the RISC, it is worth noting that in most cells including neurons over 95% of the RISC content are miRNAs. The role of miRNAs in AD has been intensively studied[77,78]. Most analyses on AD associated miRNAs are based on quantifying miRNAs from total small RNA fraction and different miRNAs have been associated with the disease[77,78]. However, we and others have found that miRNAs vary dramatically in their ability to be taken up by the RISC[28,79]. We therefore decided to subject various in vitro and in vivo systems to an analysis of RISC-bound sRNAs. We chose an Ago-RP-Seq method rather than a method involving cross-linking miRNAs with their targets, as we were not per se interested in identifying miRNA targets. Here we provide data suggesting that DISE correlates with the neurotoxicity seen in AD.

In vitro experiments confirmed a contribution of the RISC and RNAi to the toxicity of Aβ42. In addition, we found a significant association between the level of neurotoxicity and the 6mer seed viability of R-sRNAs in different models of AD. Lower 6mer seed viability correlated with higher levels of cell death. Correspondingly, a higher overall 6mer seed viability of the RISC content may protect neurons from dying. 6mer seed viability was lower in older mouse brains and in in vitro aged iPSC-derived neurons suggesting a loss of protective miRNAs with age. While our data suggest that this process is attenuated in SuperAgers, these data will need to be confirmed with larger numbers of participants once available.

There is ample literature to support the notion that miRNAs are neuroprotective in a number of disease models: In Drosophila impairing miRNA processing dramatically enhanced neurodegeneration caused by the CAG repeat gene ATXN3[80]. We previously demonstrated how toxic CAG repeat-derived sRNAs cause cell death[64,68]. Consistently, pathogenic ATXN3 with amplified CAG repeats showed strongly enhanced toxicity in HeLa cells after knockdown of Dicer[81]. Multiple publications on mice with a tissue specific knock-out of Dicer in brain cells showed that reducing Dicer triggers neuronal loss and causes behavioral abnormalities, followed by the premature death of the animals[52,53]. Moreover, mice with a tissue specific deletion of Dicer in spinal motor neurons exhibit hallmarks of spinal muscular atrophy (SMA) with signs of denervation[82]. Interestingly, mice with a brain specific knock out of Ago2 (using the same promoter: CaMKII Cre) showed no toxicity in neurons[83] suggesting that it is not the general activity of miRNAs and RNAi that is required for survival of neurons.

Conversely, increasing miRNA levels can be neuroprotective. Amyotrophic Lateral Sclerosis (ALS) disease mice treated with Enoxacin[84], showed reduced disease scores[85]. In addition, it was directly shown that miRNAs in Dicer knockout mice protect adult dopamine (DA) neurons from neurotoxicity[73] and Enoxacin treatment promoted survival of cultured DA neurons. Enoxacin could have two beneficial effects, it may increase the amount of neuroprotective miRNAs[72] and boost DNA damage repair[74].

Based on our data, we are now proposing a model in which RISC functions as a central rheostat, affecting neuronal cell death and DNA damage observed during aging and in AD, with potential implications for other neurodegenerative diseases:

In young, normal or asymptomatic individuals the RISC is filled with abundant miRNAs, most of which contain nontoxic 6mer seeds and act as protectors, preventing large amounts of small RNAs (including rRNA and tRNA fragments) that carry G and C-rich regions

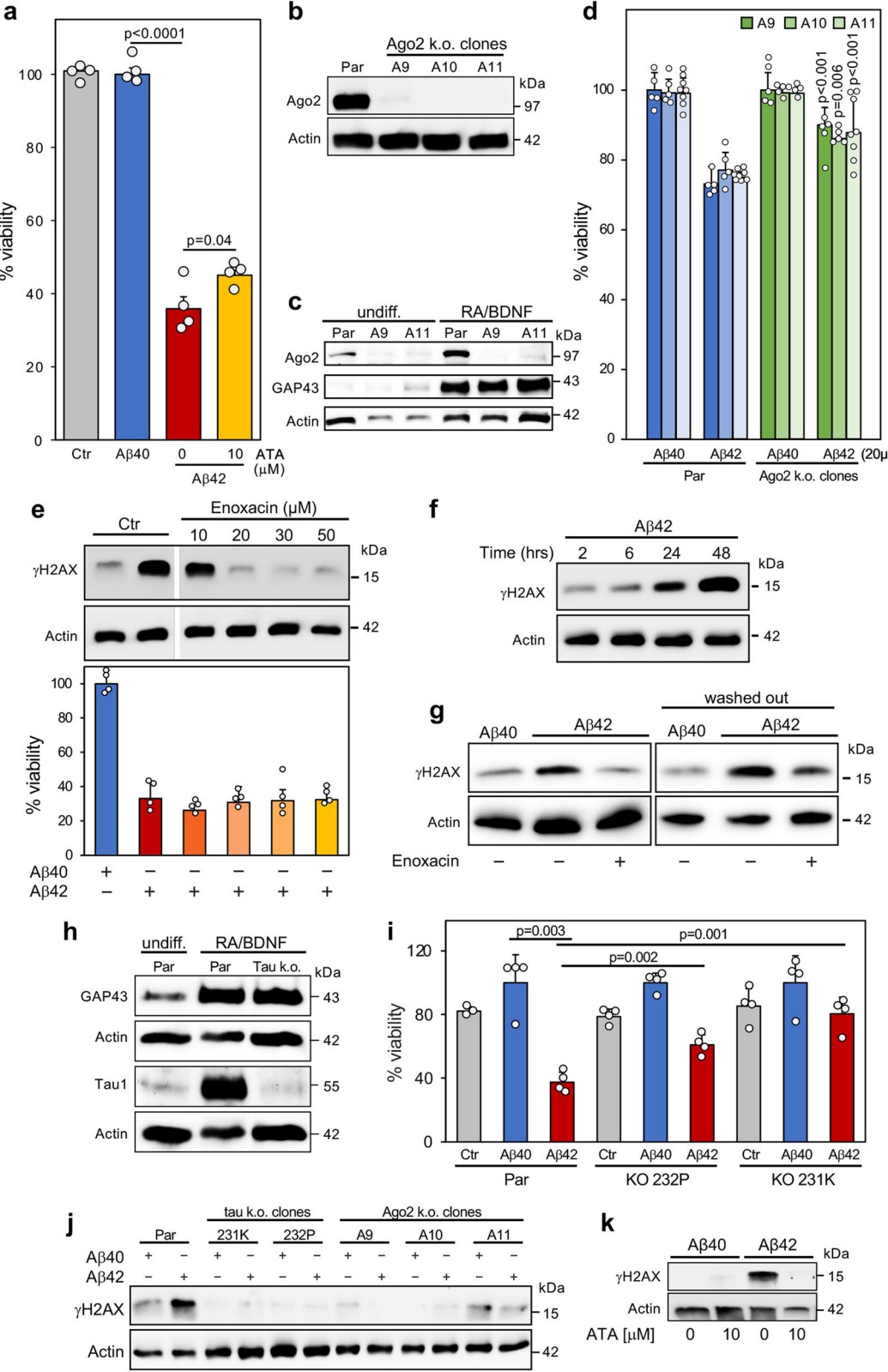

and can be toxic through DISE from entering the RISC (Fig. 7a). tRNA and rRNA fragments have been reported to be increased in the brain of AD patients[86]. miRNAs may play at least two fundamental roles in this context: First, they act as signal regulators by targeting hundreds of genes, mostly involved in development and differentiation. Second, they are expressed at high levels preventing access of toxic endogenous sRNAs to the RISC.

In AD patients once they get older Aβ oligomers and/or Tau aggregation can elicit a cellular response that causes upregulation of endogenous toxic sRNAs (miRNAs and other sRNAs including tRNA and rRNA fragments) (Fig. 7b). This may occur through a direct effect of AβOs or Tau aggregation or indirectly and these two processes could be cumulative. Mechanistically, our data suggest a moderate but significant contribution of DISE to the cell death seen in cells exposed

**Fig. 6 | Dependence of Aβ42-induced cell death and DNA damage on an active RISC. a** Viability assay of differentiated SH cells control treated for 72 hrs with either 10 μM Aβ40 or Aβ42 in the absence or presence of 10 μM ATA. Experiment was done in four technical replicates. **b** Western blot analysis of parental (Par) SH cells and three Ago2 k.o. clones. Experiments in a and b are representative of three independent biological repeats. **c** Western blot analysis of undifferentiated or 7-day differentiated SH cells and two Ago2 k.o. clones. **d** Viability assay of three Ago2 k.o. SH clones after exposure to 20 μM Aβ40 or Aβ42 for 72 hrs. Another k.o. clone was analyzed with similar results. This represents three independent experiments. Analysis of clones A9 and A10 is based on 5 technical replicates and of clone A11 on 8 technical replicates. P-values are given for the comparison of treated k.o. with parental cells in the same experiment. **e,** Top, Western blot analysis of differentiated SH cells control treated or pretreated with different concentrations of Enoxacin for 24 hrs and then treated with either 20 μM Aβ40 or Aβ42 for 24 hrs. This is representative of three independent experiments. Bottom, viability assay of the cells treated as above but for 72 hrs. This is based on 4

technical replicates. **f** Kinetics of H2AX phosphorylation of differentiated SH cells treated with 20 μM Aβ42. **g** Western blot analysis of differentiated SH cells first treated with 20 μM of Aβ40 or Aβ42 for 24 hrs and then control treated or treated with Enoxacin for 2 hours. In the sample on the right Aβ42 was washed out before an additional incubation for 2 hours. These data are representative of two independent experiments. **h** Western blot analysis of undifferentiated or 7-day differentiated parental SH cells or tau k.o. clone 231 K. **i** Viability assay of differentiated parental SH cells or two tau k.o. clones after mock treatment or treatment with 20 μM Aβ40/42. Experiment represents 4 technical replicates. This experiment was repeated with both differentiated and undifferentiated cells with similar results. **j** Western blot analysis of parental SH cells or different k.o. clones treated with 20 μM Aβ40/42 for 24 hrs. This is one of two independent experiments. **k** Western blot for γH2AX of 7 day differentiated SH cells treated with Aβ40 or Aβ42 for four hours in the absence or pretreated for 24 hrs with of 10 μM ATA. This experiment was repeated twice with different ATA concentrations. Mean with SD and two-sided Student's t-test p-values are shown (**a, d, i**).

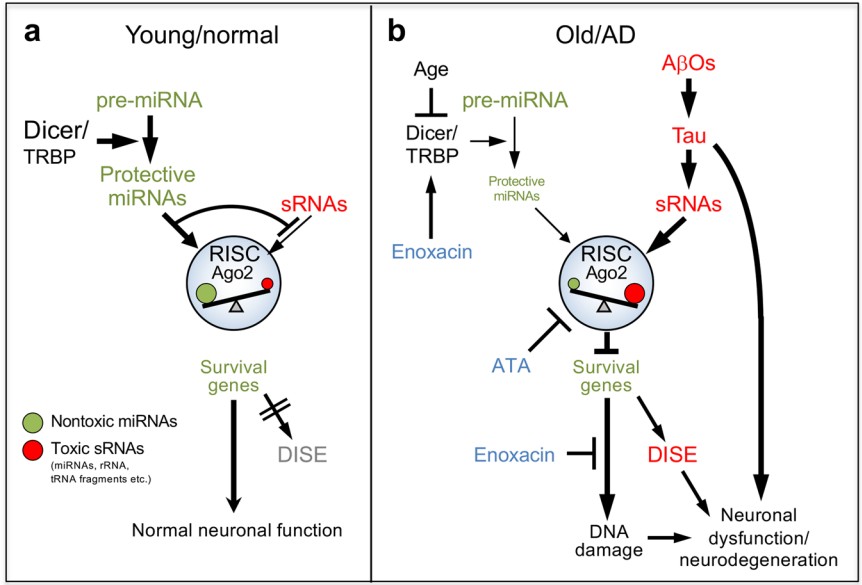

**Fig. 7 | A hypothetical model that connects DISE, DNA damage, AD, and aging. a** The proposed situation in normal or asymptomatic individuals at a young age. The ratio of nontoxic to toxic sRNAs that are bound to the RISC prevents cells from undergoing DISE. **b** The proposed situation in older and symptomatic AD patients. Dicer expression is reduced with age resulting in a gradual reduction of nontoxic miRNAs in turn causing a shift in RISC-bound sRNAs to lower 6mer seed viability

sensitizing cells to DISE and DNA damage. Tau mediates Aβ42 oligomer (AβO)-induced cell death in part through the RISC and DISE. In contrast, DNA damage induced by Aβ42/tau is dependent on RISC activity. RISC, RNA-induced silencing complex; DISE, death induced by survival gene elimination; sRNA; short RNA. Two inhibitors (blue) and their site of action are shown.

to toxic Aβ42. We saw ~10% protection from Aβ42-induced cell death in cells with ATA inhibited Ago2 or after knock-out of Ago2. We also saw a ~10% sensitization of Drosha k.o. cells lacking nontoxic miRNAs and this sensitization could be reversed by adding back some of the nontoxic miRNAs most downregulated in the k.o. cells. Although the knock-out of Ago2 or Drosha individually may lead to wide spread secondary effects, in collective these data suggest that RNAi is involved in the toxicity induced by Aβ42.

Knock-out of tau had a fundamental effect and very substantially protected cells from cell death induced by Aβ42. These findings suggest that tau is required for Aβ42 to engage the RISC and to mediate toxicity. However, the DNA damage caused by Aβ42 treatment appears to be entirely dependent on RISC engagement at least in the cell systems we used. Blocking or reversing DNA damage by Enoxacin treatment did not rescue cells from cell death suggesting the existence of pathway(s) that are parallel to DISE.

With increasing age the balance of R-sRNAs with toxic versus nontoxic 6mer seeds shifts towards lower 6mer seed viability rendering neurons susceptible to DISE inducing sRNAs. The age-dependent

loss of nontoxic miRNAs may be due to a reduced ability of aging neurons to produce enough nontoxic miRNAs. How could the amount of nontoxic miRNAs be regulated during AD and ageing? The stability of Dicer which is critical to miRNA expression in the brain is sensitive to ROS and interferons (IFNs) which are both elevated during AD[87,88] and most of its expression in the brain is lost during aging[54]. In addition, it was recently shown that the strongest expression of Drosha was detected in the cells with pyramidal neuronal morphology in the Cornu Ammonis (CA1-3) regions of the hippocampus. During AD-related stress, such as elevated ROS, the p38 MAPK pathway is activated, leading to the phosphorylation of Drosha[89]. Consequently, Drosha undergoes translocation from the nucleus to the cytosol[90]. This event is also expected to result in a loss of most miRNAs, similar to our findings in Drosha k.o. NB7 cells.

Our data point at two responses of cells to Aβ42 that involve RISC activity: Cell death/DISE and DNA damage (Fig. 7b). DISE is a combination of many cell death pathways[33] and which one is activated depends on the transcriptome of the affected cell. This is consistent with multiple cell death pathways having been implicated to

contribute to AD[2,3,91]. While cell death is a component of AD, it is thought to be a relatively late event in the collection of cascades that ultimately result in symptoms. That is why many in the field do not consider cell death as being essential. However, in all models of neuronal dysfunction, whether it is hyperactivity or neuronal silencing resulting in cognitive impairment, neuronal dysfunction may also lead to cell death[92]. In a large and unique cell type such as a post-mitotic neuron, the disease can even affect cellular substructures as shown for developing axons due to a local activation of the RNAi machinery[93]. When patients die, they have lost a substantial amount of their brain cells.

DNA damage is seen in AD and it is accelerated during aging[12,94] and mutations increase in single AD neurons[13]. Based on our data the DISE mechanism may not only contribute to neuronal cell death but also to neurodegeneration by inducing DNA damage. Future experiments will have to determine whether engagement of the DISE mechanism contributes to synaptic dysfunction and neurodegeneration and the pathology of AD.

Increase in toxic sRNAs in the context of gradual loss of nontoxic miRNAs during aging may also be at the core of other degenerative diseases that affect the brain such as Parkinson's disease (PD), Huntington's disease (HD), and ALS. It is interesting to note that one of the causes of ALS lies in the gene C9orf72[95] which contains the hexameric repeat GGGGCC. Any sRNA that contains such sequences assuming that it can enter the RISC would be expected to be highly toxic to cells as such G-rich sequences give rise to some of the most toxic sRNAs we identified (see 6merdb.org). Because the loss of nontoxic miRNAs in humans likely occurs over decades, this would explain why so many neurodegenerative diseases have an onset during adulthood with decades of symptom-free life. While the genetic modifiers and the triggers of DISE will likely be different in each of these diseases, the mechanism to protect cells from toxic sRNAs may be shared. Also, various forms of cell death have been described to occur in all of these diseases[96], however, as DISE is a combination of multiple cell death pathways depending on the transcriptome in the affected cells[26], some of these observed forms of cell death may be caused by cells undergoing DISE.

Molecular profiling data have been obtained at the genomic, transcriptomic, proteomic and metabolomic levels to further understanding of AD pathogenesis[97–100]. Our work now introduces another layer, a potential role of RISC-bound short RNAs. Our data suggest an age-dependent loss of nontoxic and potentially neuroprotective miRNAs, together with the upregulation or the presence of toxic sRNAs, alongside numerous genetic modifiers and genetic mutations found in familial and sporadic AD, may collectively contribute to the onset and progression of AD.

The overwhelming investment in AD drug discovery has been focused on two mechanisms: 1) Reducing amyloid plaque load in the brain (the hallmark of AD diagnosis; 70-80% of the effort) and 2) preventing tau phosphorylation. However, treatments aimed at reducing amyloid plaque burden have not yet resulted in an effective treatment of AD (summarized in[101]). Our data support a hypothesis that high expression of nontoxic miRNAs protects from neurodegeneration and that increasing miRNA biogenesis or blocking toxic R-sRNAs may be a viable treatment option for many neurodegenerative diseases including AD.

## Methods

### Mouse and human brain tissue
All animal experiments were approved and conducted in accordance with IACUC regulations at Northwestern University, St. Jude's Children's Hospital and University of South Florida, Tampa. 5XFAD mice were bred in house by crossing 5XFAD transgenic males (line 6799) to C57BL/6 J/SJL F1 hybrid females and genotyped as described[34]. Mice were sacrificed by $CO_2$ inhalation. One hemibrain was snap-frozen in

liquid nitrogen. Four 6- or 8-month-old female 5XFAD Rubicon[+/-] or four 5XFAD Rubicon[-/-] mice[38]; or three 3-month-old male control and three human tau45-230 transgenic mice generated using the human cDNA coding sequence for tau45-230 under the control of the Thy 1.2 promoter on a C57BL/6 J background[44] were analyzed. Young (2-month) and old (18-month) male mice (C57BL/6 J) were purchased from Jackson Laboratory.

Written informed consents were obtained from all human participants and the study was approved by the Northwestern University Institutional Review Board and in accordance with the Helsinki Declaration. We have complied with all relevant ethical regulations. Frozen brain tissue from the middle frontal gyrus of three cognitively normal elderly participants and three cognitive SuperAgers were obtained from the Northwestern University Alzheimer's Disease Center Brain Bank. Brains of cognitively normal individuals were free of neurodegenerative pathology, except for age-appropriate accumulation of plaques and tangles[102]. SuperAgers participants were enrolled in the Northwestern University SuperAging Program, were 80 years or older, were required to perform on the Reys Auditory Verbal Learning test of episodic memory equal to or better than individuals 20-30 years younger, and on tests of other cognitive domains equal to or better than same-age peers[57]. See Supplementary Table 1 for characteristics of participants.

### Reagents and antibodies
The following reagents and antibodies were used (sources given in brackets): Dibutyryl cyclic AMP (dbcAMP) (Sigma Aldrich #D0260), *all-trans* retinoic acid (RA) (Sigma Aldrich #R2625), brain-derived neurotrophic factor (BDNF) (GeminiBio #300-104 P), amyloid β-protein (Aβ42) trifluoroacetate salt (Bachem #H-8146.5000), Amyloid β-Protein (Aβ40) trifluoroacetate salt (Bachem #H-1194.5000), aurintricarboxylic acid (ATA) (Sigma Aldrich #189400), Enoxacin (Millipore Sigma # 557305), Opti-MEM (Thermo Fisher Scientific #31985088), Lipofectamine RNAimax (Thermofisher Scientific #56532), Lipofectamine 3000 (Thermofisher Scientific #100022052), 1,1,1,3,3,3-Hexafluoro-2-Propanol (HFIP) (Sigma Aldrich #105228), argonaute-2 rabbit pAb (Abcam #ab32381, 1:1000), tyrosine hydroxylase rabbit pAb (TH) (Cell Signaling #2792, 1:1000), GAP43 (D9C8) rabbit mAb (Cell Signaling #8945, 1:1000), tau1 mouse mAb, clone PC1C6 (Millipore Sigma #MAB3420, 5 μg/ml), Drosha (D28B1) rabbit mAb (Cell Signaling #3364, 1:1000), Dicer (D38E7) rabbit mAb (Cell Signaling #5362, 1:1000), goat anti-rabbit secondary Ab (Southern Biotech #4030-05, 1:10,000), goat anti-mouse secondary Ab (Southern Biotech #1070-05, 1:10,000), HRP-conjugated β-actin (C4) mouse mAb (Santa Cruz Biotechnology #Sc-47778, 1:10,000), phospho-histone H2A.X (Ser139) (20E3) rabbit mAb (Cell Signaling #9718, 1:1000), neuronal class III β tubulin (TUJ1, Millipore #T8660, 1:1000) with secondary Ab Alexafluor goat anti-mouse IgG2b 488 (Invitrogen #A-21141, 1:1000), glutamatergic neuronal marker vesicular glutamate transporter (Vglut1, Synaptic Systems #135 304, 1:500) with secondary Ab Alexafluor 555 goat anti-guinea pig Invitrogen #A-21435, 1:1000), LMX1A (rabbit pAb, Millipore #AB10533, 1:1000 IF), tyrosine hydroxylase (pAb, Millipore #657012, 1:1000 IF), HNF-3β RY7 (FOXA2) (mouse mAb, Santa Cruz # sc-101060, 1:100 IF), TUJ1 (mouse mAb, Covance #MMS-435P, 1:2000) used for WB or IF (used in Supplementary Fig. 4) with secondary Abs anti-mouse or rabbit IgG conjugated Alexa 488 (1:400) or Alexa 568 (1:200) (Life Technologies).

### Aβ peptide preparation
Solubilization, stock preparation and oligomerization of Aβ42 was performed as previously described[103,104] with some additional modifications. Briefly, lyophilized Aβ42 was allowed to equilibrate at room temperature for 30 min and resuspended in ice cold HFIP under fume hood to obtain 1 mM solution. To disintegrate previously formed random oligomers and achieve complete monomerization, solution

was vortexed vigorously for 1 min and kept at RT for 2 hrs. After monomerization, peptide solution is aliquoted in desired volumes into 1.5 ml microcentrifuge tubes, air dried overnight and then transferred to SpeedVac for another 2 hrs to remove traces of HFIP. Complete removal of HFIP and quality of monomer formation was validated by the formation of clear peptide film observed at the bottom of the tubes. Tubes containing peptide films were parafilmed and stored at -80 °C for further use. For oligomerization, tube containing peptide film was dissolved in DMSO at 5 mM, vortexed for 30 sec and sonicated in the water bath for 10 min for complete suspension. Peptide was further diluted to 100 μM, vortexed, briefly centrifuged and kept at 4 °C for 24 hrs for oligomer formation. Media to dilute peptide depended on the experimental condition and lacked FBS and phenol red components. After 24 hrs low molecular weight soluble peptide oligomers were retrieved by centrifugation at 14,000 g for 10 min and applied at various concentrations in toxicity assays. To control for HFIP toxicity Aβ40 was prepared in the same manner as Aβ42.

## Cell culture

Human neuroblastoma cell lines; SH-SY5Y (# CRL-2266™) (SH cells) were purchased from ATCC and NB7 were kindly provided by Dr. Jill Lahti (St. Jude Children's Research Hospital, Memphis, TN)[37]. SH-SY5Y cells (derived from a bone marrow biopsy of a metastatic neuroblastoma of a 4-year-old female) were cultured in Dulbecco's Eagle's medium (DMEM) (Corning # 10-013-CM) and NB7 cells in RPMI 1640 medium (Corning # 10-040-CM), both supplemented with 10% heat inactivated fetal bovine serum (FBS) (Sigma-Aldrich # 14009 C, 1% L-glutamine (Corning # 25-005-Cl) and 1% penicillin/streptomycin Corning # 30-002-CI). Cells were maintained at 37 °C in saturated humidity atmosphere of 95% air and 5% $CO_2$. The two tau k.o. clones 232 P and 231 K were obtained from Dr. Paolo Paganetti (Laboratory for Biomedical Neurosciences, Ente Cantonale Ospedaliero, Torricella-Taverne, Switzerland). These cells were cultured in complete DMEM media supplemented with 1% non-essential Amino Acids (NEAA) (Cambrex Bio Science # 13-114E).

## Generation of k.o. cells

NB7 Drosha k.o. cells were generated using the Alt-R CRISPR-Cas9 System (IDT DNA Technologies) following the manufacturer's protocol with the following modifications. A paired set of sgRNAs, one targeting in exon 2 (IDT predesigned gRNA Hs.Cas9.DROSHA.1.AB GUACAAA-GUCUGGUCGUGGA) and one targeting downstream of exon 32 (IDT custom design AUAACUUGAUGAACAGCCAC) were annealed with the Alt-R Crispr Cas9 tracrRNA 5′ ATTP 550 (#1077024) and mixed 1:1. The sRNA mix (22 pmol) was incubated with 18 pmol Alt-R Cas9 Nuclease V3 (#1081058) diluted in Resuspension Buffer R (Invitrogen #MPK10025). The Invitrogen Neon Transfection System was used to transfect the Cas9-sgRNA complex into $2 \times 10^5$ NB7 cells in 200 ml Resuspension Buffer R with Cas9 electroporation enhancer (1200 V, 30 ms width, 1 pulse). The next day single PE+ cells were sorted into 96-well plates and cultured for ~ 6 weeks. SH-SY5Y Ago2 k.o. pool was generated by Synthego using the guide RNA sequence: UAAUUU-GAUUGUUCUCCCGG. Single cells were sorted into 96-well plates with 50% conditioned media by FACS. Knockout clones were confirmed by western blotting.

## Cell growth and viability assessment

Transfection of all cell lines (SH, NB7 and their Drosha k.o. clones) was performed in TPP 96 well plates (Millipore Sigma # 92696) in 250 μl final volume. Briefly, 50 μl Opt-MEM containing sRNA-lipofectamine RNAiMax (NB7) or sRNA-lipofectamine 3000 (SH) complex was forward transfected to 200 μl cells containing 3,000 NB7 cells/well or 6,000 SH cells/well. The optimized amount of lipofectamine RNAiMax or lipofectamine 3000 for both cell lines was 0.3 μl. Short (s)RNAs used to transfect these cells were sNT1, sGGCAGU, sGGGGGC or sCAG

(sequences all with their guide strands modified by 2′-O-methylation of their positions 1 and 2 as previously reported[25,64]). Following transfection, cell growth was monitored for at least 160 hrs in the IncuCyte Zoom live-cell imaging system (Essen Bioscience) with a 10X objective. The cell confluency curves were generated using IncuCyte Zoom software (version 2016A). To determine the presence of metabolically active cells after various treatments, ATP quantification was performed using Cell/Titer-Glo Luminescent Cell Viability Assay Kit (Promega # G7571). Following Aβ40/Aβ42 or ATA treatment in 96 black well plate, an equal volume of Cell Titer-Glow reagent was added to the amount of cell culture media in the well. The contents were mixed in an orbital shaker for 2 min to induce cell lysis and incubated for another 10 min to stabilize the luminescent signal. Subsequently, luminescence signal was recorded using microplate reader (BioTek Cytation 5). A small number of data points were eliminated from analysis due to technical issues.

## Differentiation of SH cells

SH cells were differentiated into neuronal-like phenotypes by two methods as described previously, one by using dbcAMP[105] and other by the combination of retinoic acid (RA) and brain-derived neurotrophic factor (BDNF)[106,107]. Briefly, SH cells maintained in DMEM media were seeded at a density of 10,000 cells/well in 96 black well plate. After 24 hrs, DMEM media was removed, and cells were differentiated with 1 mM dbcAMP in Nb-active no phenol red (BrainBits # NbActiv4-PR) media for 7 days. At day 5 of differentiation, undifferentiated cells were again seeded at same density in parallel. At day 7, when cells were differentiated, they were treated with 20 μM (unless otherwise noted) Aβ40/Aβ42 for 48 hrs and ATP cell viability was measured as described above. To achieve neuron-like differentiation with RA/BDNF, cells were first pre-differentiated with 10 μM RA in complete DMEM media for 5 days and then treated with 50 ng/ml BDNF in the same media that lacked FBS and phenol red for an additional 2 days. Seven days differentiated neuron-like cells were then used to assess toxicity of Aβ40/Aβ42 in all experiments involving differentiated SH cells.

## Aurintricarboxylic acid and Enoxacin treatment

Stock solution of ATA was prepared in 0.1 M NaOH and Enoxacin in 1 M NaOH. Prior to sRNA transfection or treatment with Aβ42 cells were pretreated with indicated concentrations of ATA for 2-24 hrs and its presence was continued throughout the analyses. In the case of Enoxacin cells were only pretreated for 24 hrs.

## RNA extraction, reverse transcription, and quantitative real time PCR

Monolayer of $1.0 \times 10^6$ cells grown in 6 well plate was lysed in 500 μl/well QIAzol Lysis Reagent (Qiagen #79306) and RNA extraction was performed with QIAzol miRNeasy Mini Kit (Qiagen #217004) according to the manufacturer's instruction. All RNA samples were subjected to DNAse digestion using RNAse-free DNAse set (Qiagen #79256) and eluted in 35 μl ultra-pure $H_2O$. RNA quality control was performed with Nanodrop 2000c spectrophotometer. Subsequently, 100 ng RNA was reverse transcribed using High-Capacity cDNA Reverse Transcription Kit (Applied Biosystems # 4368813) in 15 μl. cDNA was further diluted in ultra-pure $H_2O$ to 45 μl, and qPCR was performed with TaqMan Universal PCR Master Mix (Thermo Fisher scientific #4324020) using 5 μl cDNA in a 20 μl total volume. Assays were run on 7500 Real Time PCR System (Applied Biosystems). Small nucleolar RNA (Z30) (Thermo Fisher Scientific, #001092) that have been reported to show good abundance and relatively stable expression was used as an endogenous control. Primers used to quantify expression levels of miRNAs in Parental and Drosha k.o. clones were hsa-miR-21 (Thermo Fisher Scientific, #0003970), Let-7a (Thermo Fisher Scientific # 000377), hsa-miR-221 (Thermo Fisher Scientific #000524) and hsa-miR-182 (Thermo

Fisher Scientific #002334). Relative expression of miRNAs analyzed in triplicates were normalized to the levels of Z30 and expressed as $2^{-\Delta\Delta Ct}$.

## Western blot analysis

Pellets containing ~$1.0\times10^6$ cells were lysed in 250 µl RIPA buffer (150 mM NaCl, 10 mM Tris HCl pH 7.2, 1% SDS, 1% Triton X-100, 1% deoxycholate, 5 mM EDTA) supplemented with 1 mM PMSF and protease inhibitor cocktail (Roche #11836170001). To ensure complete lysis, cells were kept on ice for 30 min, vortexed occasionally and sonicated. Lysed cells were then boiled for 10 min at 95 °C and centrifuged (14,000 g at 4 °C) for 15 min. Protein quantification was performed with DC Protein Assay Reagent (BIO RAD) in an iMark Microplate Reader (BIO RAD) at 750 nm. 20 µg protein per lane was run on SDS-polyacrylamide gel electrophoresis (SDS-PAGE) (BIO-RAD #1610158) at 40 mA for ~1 h. Resolved proteins were transferred to Nitrocellulose Blotting Membrane (Healthcare #10600016) at 250 mA for 2 hrs on ice. Transferred membranes were then blocked with 5% non-fat dry milk for 1 hr at room temperature in TBST (Tris-buffered saline, 0.1% Tween 20) and immunoblotted with various antibodies overnight (1:1000 in 3% BSA in TBST). Following primary antibody incubation, membranes were washed with TBST and probed with secondary goat-anti-rabbit or mouse IgG HRP (1:5000 in 1% milk for 1 hr at RT). Finally, membranes were washed and developed with Super Signal West Dura Luminol/Enhancer (Thermo Scientific #1859025).

Western blot analysis of iPSC-derived midbrain dopamine neurons: Cultured cells were harvested at different time points as indicated in the figures and lysed in 1% Triton X-100 buffer containing protease inhibitor cocktail (Roche diagnostics, # 11-836-170-001), 1 mM PMSF, 50 mM NaF, 2 mM sodium orthovanadate by homogenization and incubation on ice for 30 minutes. The protein concentration was determined by using micro-BCA kit (ThermoFisher, # 23235) and 40 µg of lysates were loaded onto 10% Tris/glycine PAGE gels followed by transfer onto PVDF membranes (EMD Millipore, # IPFL00010) at 30 V for 1 hr). Membranes were post-fixed in 0.4% paraformaldehyde, washed in milliQ water and then blocked in 1:1 TBS: odyssey blocking buffer (Licor # P/N 927-40003) for 1 hr at room temperature. The membrane was incubated with primary antibodies diluted in 1:1 ratio 0.2% TBS-tween and odyssey blocking buffer overnight at 4°C. The following day the membrane was washed with 0.2% TBS-Tween and incubated with secondary antibodies for 1 hr. Anti-rabbit IgG conjugated to IRDye800 and anti-mouse IgG conjugated to Alexa 680 were used for simultaneous detection of both channels on the same blot. The blot was washed as before and scanned on an odyssey imaging system. The western blots were analyzed using Image Studio software (Licor) to quantify band intensities. All uncropped blots are shown in Supplementary Fig. 10.

## TUNEL staining of mouse brains

Brains of the indicated genotypes were harvested from mice following transcardic perfusion with 4% PFA in PBS. Harvested brains were immediately placed in 4% PFA and fixed overnight at 4°C. Brains were then briefly washed with cold PBS and transferred to 30% sucrose for cryoprotection. Brains were oriented following sagittal segmentation in OCT, sectioned, and mounted on glass slides. TUNEL staining was performed using the 594-TUNEL Assay Kit (Cell Signaling #48513) per the manufacturer's instructions. In brief, slides were rinsed in PBS to remove residual OCT medium and then permeabilized in 5% BSA/0.2% TX-100 in PBS for 30 min. Slides were washed two times in PBS then incubated for 5 min in TUNEL equilibration buffer. Next, slides were incubated for 2 hrs at 37 C in TUNEL reaction mix in a humidified, dark environment. Slides were then washed 3 times in BSA/TX PBS solution above. Finally, coverslips were applied using Prolong Diamond Antifade Mountant with DAPI (Invitrogen P36931).

Slides were then imaged on a Nikon Ti-E epifluorescence microscope with Andor Clara ECCD camera and processed using Nikon Advanced Research (NIS) Elements software. Quantification of TUNEL+ cells was achieved by automatically defining and separately masking TUNEL signal and the DAPI signal using NIS Elements. TUNEL+ cells were defined where TUNEL signal was observed in the DAPI masked area using automated analysis. Automated results were subsequently verified by manual counting. TUNEL signal that was observed outside of the DAPI masked area was not included in this analysis for both automated and manual counting, although may represent additional cell death in cells that no longer stain DAPI positive.

## iPSC-derived excitatory forebrain neurons from AD patients

Generation of iPSC lines from AD patients (AD1: AG5810-F [APOE3/4]; AD2: AG011414 [APOE3/4]) and healthy controls (Control 1 NUAD0635 [PBMC], Control 2: AG02261 [APOE3]) were previously described and obtained from Coriell Institute for Medical Research[108] (AG5810, AG011414, AG002261) or provided by Northwestern University's Mesulam Center for Cognitive Neurology and Alzheimer's Disease (NUAD0635). iPSCs were differentiated to forebrain excitatory neurons via lentiviral-mediated overexpression of *neurogenin-2*, as previously described[109] with minor modifications. Stem cells were maintained in mTeSR1 media (Stemcell Technologies) on Matrigel coated dishes (Corning). Upon start of differentiation, cells were single cell dissociated using Accutase (Millipore) and resuspended in mTeSR1 with 10 µM ROCK Inhibitor Y-27632 with lentiviruses encoding rrTA and pTetO-Ngn2-puro (generated by the Northwestern University Skin Biology & Diseases Resource-Based Center. The following day, medium was changed to KO-DMEM (Thermo Fisher Scientific), 1X MEM nonessential amino acids, 1X Glutamax, and 0.1% 2-mercaptoethanol, supplemented with 10 µM SB431542 (Stemgent), 100 nM LDN193189 (Stemgent), 2 µM XAV939 (Tocris Bioscience, Bristol, UK), and 3 µg/ml doxycycline (Sigma). Gradually, medium was changed over 2 days to neural induction medium, DMEM/F-12 containing MEM nonessential amino acids, Glutamax, N-2 (Thermo Fisher Scientific), D-glucose (Sigma), 2 µg/ml heparin sulfate (Sigma) supplemented with 3 µg/ml doxycycline and 2 µg/ml puromycin. Induced neurons were replated onto 6-well tissue culture plates at 450k cells/well on pre-coated with poly-L-orthinine (Sigma), 4 µg/ml laminin (Roche Basel, Switzerland), and 2 µg/ml fibronectin (Sigma), cultured with neuronal maturation medium (BrainPhys Basal Medium [Stemcell Technologies], B-27 and N-2 supplements [Thermo Fisher Scientific], MEM nonessential amino acids, and Glutamax, supplemented with 3 µg/ml doxycycline and 10 ng/ml BDNF [R&D systems]). Post-plating, cells were treated with 3 µM Ara-C for 48 hrs to eliminate proliferating cells. Half of the medium was exchanged every 2 to 3 days until cell collection.

## iPSC-derived midbrain dopamine neurons used for in vitro aging study

Human iPSC from a previously characterized healthy control (line 2135 or C3 in[110]) were cultured in mTeSR1 media (Stemcell Technologies) on vitronectin coated dishes. iPSCs were dissociated into single cells by accutase treatment and seeded at 5,000 cells per cm² in a Matrigel coated dish. Differentiation was carried out as described[110] using dual SMAD inhibition followed by stimulation of sonic hedgehog and canonical WNT signaling. Cells were passaged en bloc at day 15 and then manually passed onto 10 cm dishes coated with poly-D-lysine at 66 µg/ml (Sigma # P1149) and laminin at 5 µg/ml (Roche # 11243217001). Cells were cultured in the presence of differentiation factors for 40 days, and then maintained in Neurobasal media (Life Technologies) supplemented with Neurocult SM1 (Stemcell Technologies) and 1% penicillin/streptomycin (Life Technologies) until harvesting up to day 190.

To characterize iPSC-derived neurons by immunocytochemistry, differentiated midbrain neurons were fixed in 4% paraformaldehyde in phosphate buffered saline (Life Technologies) for 20 min followed and

permeabilized / blocked in PBS with 0.3% Triton X-100 containing 2% bovine serum albumin and 3% normal goat serum for 30 min. Primary antibodies were added and incubated overnight at 4°C in blocking buffer. Cells were then incubated in anti-mouse or rabbit IgG conjugated Alexa 488 (1:400) or Alexa 568 (1:200) (Life Technologies) for 1 hr, washed, then mounted in 10 µl of 4,6-diamidino-2-phenylindole dihydrochloride (DAPI)-containing Fluoromount G (Southern Biotech, #0100-20). Coverslips were analyzed by microscopy using a Leica confocal microscope (Leica TCS SPE laser at 25-50% power; CTR4000 / DMI4000B microscope) through a 10 µm section (z-series at 1 µm per section).

**Ago pull-down and subsequent small RNA-seq (Ago-RP-Seq)**
Ago pull down of the samples used in the study was performed as previously described[28]. Briefly, frozen mouse brain pieces (~ 150 mg, cortex and hippocampus combined) and the cell pellets obtained from iPSC-derived neurons (10^6), RA/BDNF differentiated SH cells (1.5 ×10^6), NB7 parental and Drosha k.o. clones (10^6), NB7 cells transfected with sRNAs (1.5×10^6) were lysed in 1 ml NP40 lysis buffer [50 mM Tris pH 7.5, 150 mM NaCl, 5 mM EDTA, 0.5% NP-40, 10% (v/v) glycerol, 1 mM NaF; supplemented with 1:200 EDTA-free protease inhibitors (Millipore #539134) and 1:1000 RNaisin Plus (Promega #N2615) before use]. For complete lysis, cells were kept on ice for 15 min, vortexed occasionally, and then centrifuged at 20,000 g for 20 min. In the case of brain tissue samples, lysates were prepared by homogenizing tissues several times using Dounce Tissue Grinder in the same lysis buffer (Duran Wheaton Kimble # 357538). Pull down of Ago proteins (Ago 1-4) was performed by incubating 500 µg of Flag-GST-T6B peptide[111] (Sigma #M8823) with 80 µl of anti-Flag M2 Magnetic beads (Sigma #M8823) at 4 °C for 3 hrs in a rotor. Subsequently, beads were washed three times and resuspended in 1 ml NP40 lysis buffer. To confirm pull down of Ago proteins, 50 µl aliquot containing beads were mixed with 2x SDS-PAGE sample buffer and run on 10% SDS-PAGE gel for western blot analysis by anti-Ago2 antibody. After validation, remainder of the beads was used for RNA extraction using Trizol reagent (Ambion #15596018). Subsequently, RNA pellet was resuspended in 20 µl ultra-pure H$_2$O and 10 µl was kept for RNA visualization purposes while another 10 µl was used for small RNA library preparation using Illumina primers (RRID:SCR_010233) as described previously[112]. Briefly, RNA was ligated with 3′-adenylated adapters using T4 RNA Ligase 2 (NEB #MO351) for 4 hrs at 16 °C. Product was ethanol precipitated and run on 15% Urea-PAGE gel. To determine correct position of the ligated product in the gel, radiolabeled (^32P) unligated and ligated size markers (19-35nt) were also run in parallel. Ligated product was then eluted from the gel and prepared for 5′ adaptor ligation. Ligation of 5′ adaptor to already 3′ ligated RNA product was performed with T4 RNA ligase 1 (NEB, #EL0021) at 37 °C for 1 h. The product was again run on 12% Urea-PAGE gel, extracted and reverse transcribed using Superscript III reverse transcriptase (Invitrogen #18080–044). Enrichment of product was performed by amplifying cDNA with multiple rounds of few cycles PCR. Quality of cDNA was validated by bioanalyzer analysis with a single peak at ~157 bp length. Sequencing of cDNA was performed on Illumina Hi-Seq 4000. The sequence information on size markers, 3′ and 5′ adaptors, RT primers, and PCR primers were previously published[113].

**6mer seed viability (SPOROS) analysis**
The SPOROS pipeline[39] was used to generate 6mer seed viability graphs, 6mer seed viability plots and 6mer seed nucleotide composition analyses. Briefly small RNAseq data sets in the form of raw fastq files were de-multiplexed, trimmed, cleaned and then compiled into a single read count table. After removing rare reads as described[39], the remaining reads were BLASTed against lists of small RNAs from either mouse or human. Reads that matched artificial sequences in the database were removed and final raw read count table (rawCounts)

was generated. The raw read count table was used to generate two read count tables; one normalized read count table (normCounts) that contained 1 million reads per sample, and the other normalized and differentially expressed read count table (differential). After annotating each count table with 6mer seed, average 6mer seed viability, miRNA, and RNA world data, various 6mer seed viability analyses were done. 6mer seed viability graphs were generated by adding and collapsing all rows that contained same 6mer seed and RNA type and aggregating all the read counts within that 6mer seed viability in 1% bin size. Graphs were prepared in Excel with smoothing line function on. Shown in the graphs are the peaks of normCounts plotted against % 6mer seed viability. When determining the average 6mer seed viability or 6mer seed composition analysis each row counts were first normalized by 1000 to ease computational complexity and the columns containing 6mer seeds and average 6mer seed viability were expanded based on the normalized read counts. Table containing frequency distribution of average 6mer viability for various groups were then used to generate basic boxplot (StatPlus (v.7.5)). Statistical analysis was done with Kruskal-Wallis median test when comparing groups. Weblogo plots were generated to show average 6mer seed composition in seed positions 1-6. The differential expression analysis between two groups was done by taking significantly (p-value < 0.05) expressed reads for each row between two groups and expressing them in delta read counts (perturbed sample-control sample). The frequency of average 6mer seed viability or 6mer seeds were then expanded based on the delta read counts to get the 6mer seed viability graph, 6mer seed viability plots, or 6mer seed compositions as described above.

**Statistical analyses**
Two-sided Student's t-tests were performed in Microsoft Excel. Two-way analyses of variance were evaluated using STATA version 14.0. STATA was also used for comparing differences in polynomial distributions as described previously[24]. All other statistical analyses were conducted in R version 4.0.0.

**Reporting summary**
Further information on research design is available in the Nature Portfolio Reporting Summary linked to this article.

## Data availability
The RNA seq data generated in this study have been deposited in the GEO database under accession code GSE213138. All data used to generate graphs in this study are provided in the Supplementary Information and Source Data files. Source data are provided with this paper.

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

## Acknowledgements

We would like to thank Drs. William Klein and Ruth Itzhaki for helpful discussions and input early in the project Dr. Paolo Paganelli for proving tau knock-out SH-SY5Y cells and Monal Patel for help with Western blotting. This work was supported by National Institutes of Health grants R01NS090993 (to A.F.), R01AG030142 (to R.V.), R35CA231620 (to D.R.G.), R35CA197450 (to M.E.P), R01AG045571, R56AG045571, R01AG067781, U19AG073153, P30AG072977, P30AG13854 (to E.R. and C.G.), R01NS124783, and L40CA231423 (to B.L.H). The content is solely the responsibility of the authors and does not necessarily represent the official views of the National Institutes of Health. Further support was provided by the McKnight Brain Research Foundation (to E.R.).

## Author contributions

M.P. designed experiments, analyzed data, and conceptualized the work. B.P. designed, performed, and analyzed most of the in vitro experiments and B.P. and S.-Y.J. performed the Ago-RP-Seq experiments. C.P.M. and A.R. stained and imaged mouse brains. AHK generated and characterized the Drosha k.o. NB7 cells. E.B. developed SPOROS and E.B. and B.P. performed bioinformatics analyses. K.F. and A.A. generated and cultured human iPSC-derived neurons. J.K. and J.M. provided iPSC-derived neurons and their characterizing data. A.M. provided conceptual input. E.R. and C.G. provided the CN and SuperAger brain samples. A.F. provided brains from tau transgenic mice. B.H. and D.G. provided the brains of 5XFAD and 5XFAD Rubicon k.o. mice and B.H. performed the TUNEL staining of these brains. K.S. and R.V. provided training and input on working with AD brains. M.P. and B.P. wrote, and R.V., A.H.K., B.P., and M.P. edited the manuscript. All authors read and approved the final manuscript.

## Competing interests

The authors declare no competing interests.

## Additional information

¹Department of Medicine/Division Hematology/Oncology, Feinberg School of Medicine, Northwestern University, Chicago, IL 60611, USA. ²USF Health Byrd Alzheimer's Center and Neuroscience Institute; Department of Molecular Medicine, Morsani College of Medicine, Tampa, FL 33613, USA. ³Department of

Biochemistry and Molecular Genetics, Feinberg School of Medicine, Northwestern University, Chicago, IL 60611, USA. [4]Department of Preventive Medicine/ Division of Biostatistics, Feinberg School of Medicine, Northwestern University, Chicago, IL 60611, USA. [5]Davee Department of Neurology, Feinberg School of Medicine, Northwestern University, Chicago, IL 60611, USA. [6]Mesulam Center for Cognitive Neurology and Alzheimer's Disease, Feinberg School of Medicine, Northwestern University, Chicago, IL 60611, USA. [7]Department of Psychiatry and Behavioral Sciences, Feinberg School of Medicine, Northwestern University, Chicago, IL 60611, USA. [8]Department of Cell and Developmental Biology, Feinberg School of Medicine, Northwestern University, Chicago, IL 60611, USA. [9]Department of Immunology, St. Jude Children's Research Hospital, Memphis, TN 38105, USA. [10]Present address: Ministry of Food and Drug Safety, Pharmaceutical Safety Bureau, Pharmaceutical Policy Division 187, Osongsaengmyeong 2-ro, Osong-eup, Heungdeok-gu, Cheongju-si, Chungcheongbuk-do, Republic of Korea. [11]Present address: Healthy Aging & Alzheimer's Research Care (HAARC) Center, Department of Neurology, The University of Chicago, Chicago, IL 60637, USA. ✉e-mail: m-peter@northwestern.edu

