## [Peer Review File · Nature Communications]

Death Induced by Survival gene Elimination (DISE) is correlated with neurotoxicity in Alzheimer's disease and agingREVIEWER COMMENTS

Reviewer #1 (Remarks to the Author):

The manuscript builds on the applications of the authors' published method (SPOROS) used to identify sRNA containing 6-mer seeds bound to RISC that may be toxic in cancer and Alzheimer's disease. In this manuscript, the authors utilize mouse models as well as postmortem samples of AD and ageing to validate the results from their previous study. The research identifies differentially expressed miRNA that range from protective to toxic based on their seed viability. The conclusions derived focus on the role of RISC-bound sRNA containing toxic 6-mer seeds that increase with ageing and trigger DNA damage and cell death pathways, while the protective miRNA decrease.

Major points

Definition of "seed viability" is confusing in the first paragraph of results. Does it refer to the cell viability in response to exposure to sRNAs? Or does it refer to the effectiveness of the sRNA in inducing DISE?

There is an error in reporting the average age of SuperAgers versus controls in the results and figure legend. The significant difference in seed viability and toxic nucleotide abundance in fig 4d is unconvincing.

It is unclear or not addressed whether the increase in toxic sRNA is a direct effect of Abeta and tau aggregation in AD or an indirect consequence. The effect of increase in toxic sRNA and Abeta toxicity on cell viability appears to be cumulative rather than interdependent. Alternatively, the mechanism by which the increase of toxic sRNA, including fragmented rRNA or tRNA, is achieved can be discussed in more detail in context of AD to improve the significance of the results obtained in this study. Further, it can also be discussed what kind of cell death pathways are triggered through DISE and how.

Where is the data to support the blockage of DNA damage in Abeta treated cells upon enoxacin treatment?

Minor points

Labelling of blue vs red in seed viability graphs is not always clear (Fig1, Fig4c,d).

Language and punctuation can be improved for clarity.

Reviewer #2 (Remarks to the Author):

In this manuscript, the authors present data to suggest that there could be a contribution of DISE to amyloid beta toxicity, using cell lines, mouse models and analysis in aged humans. They show data that there is a correlation between differentially expressed R-sRNAs and abeta-42 treatment in Sy5Y cells, they show there is a correlative shift of 6mer seed viability in the 5xFAD mouse if they have the Rubicon homozygous background; they show iPSC derived neurons compared to AD have a shift in R-sRNAs; they show a shift in older mice, iPSCs, blood cells in young, old vs superagers. They also show that drosha knockdown of cells treated with a-beta42 have a viability shift and a change in R-sRNAs, and they do experiments with Ago2, Enoxacin, and tau, to provide data of some effect of loss of Ago2 on amyloid beta toxicity.

This seems a really interesting pathway, and overall the authors have performed a large amount of work. But an overall issue, which they clearly admit, is that the effect is modest and most of their findings are correlative. So in the end, the reader is simply left with considering that this pathway appears not very important or impactful for a-beta42 toxicity in Alzheimer's disease.

The experiments of Figures 1-4 are correlative, and the effects in any case are modest, so the data are not very compelling that if indeed this pathway is contributing, it is not playing much of a role.

They try to provide evidence in Figure 5 that specific miRNAs are the most important non-toxic ones in the cells, but one wonders if any miRNA would have the same effect, and the data would be more compelling if they defined target mRNAs that are critical (do the targets function in the pathways of death predicted). In figure 6, the effects of Ago 2 are more considerable, but effects of ATA are rather modest (already previously shown on primary neurons). The most compelling data they present is on DNA damage, but this is rather a side point.

Also knockdown of drosha and ago2 may have broader effects to impact many genes, so a specific effect on DISE is hard to claim.

Other comments:

For all the graphs, the individual data points should be shown. It is also unclear in a figure like Figure 1e, how many independent experiments do those data represent?

Figure 2: The rationale for the rubicon mutant in mice in figure 2 is not clear. They cannot see differences without it, and it should only be impacting autophagy so it is confusing.

Figure 5: One wonders about including a control miRNA that is not different or changed. So a random real miRNA. Would addition of any expressed miRNA have the same effect?

Figure 6: at least how it is presented, the difference with gamma H2AX is very striking in this figure. That seems a strong effect.

Reviewer #3 (Remarks to the Author):

This is a very interesting manuscript that approaches microRNA from a novel angle. They recently showed that RISC bound sRNAs containing hexameric seeds can alter RISC function and cause toxicity. They found that the ratio of miRs with toxic vs. nontoxic seeds determines the sensitivity to a type of cell death, termed DISE. In the current manuscript they have examined the effects of A β on the types of miR bound to the RISC complex and the resulting sensitivity to DISE. The authors make a compelling case that the pattern of miRs containing hexameric repeats changes in response to A β and stimulate DISE. The manuscript is interesting and good, but would benefit from some additions.

1) While Figure 2C provides a diagram of how hexamer ratios might affect cell viability, I think the manuscript would benefit from a diagram in Figure 1 demonstrating how hexameric seeds in miRs act on DICER to alter its function. They could consider simply improving figure 2C and moving it to Figure 1.

2) Fig 2: When discussing the effects of Rubicon KO on 5xFAD, the authors should show the resulting tau pathology and microgliosis in the manuscript – perhaps as supplemental figures.

3) Fig. 4: The authors should also determine whether DICER transitions to the insoluble

fraction with aging, particularly in the AD models. They can also examine human publicly available proteomics data, such as that from Seyfreid's Nat Med paper to examine human data.

4) Fig. 6: I agree that the data show the DNA damage is a result of Abeta tox rather than mediating it. However, the authors should point out that use of enoxacin at 20 uM likely yields extensive off-target activity which complicates interpretation.

5) Do the hexameric repeats produced by C9orf72 affect DISE?

Reviewer #1 (Remarks to the Author):

Major points

Definition of "seed viability" is confusing in the first paragraph of results. Does it refer to the cell viability in response to exposure to sRNAs? Or does it refer to the effectiveness of the sRNA in inducing DISE?

We have added a clarifying sentence.

There is an error in reporting the average age of SuperAgers versus controls in the results and figure legend. The significant difference in seed viability and toxic nucleotide abundance in fig 4d is unconvincing.

We have fixed the error.

We agree that the results with the SAs are preliminary (as stated in the text). Getting access to this precious material was difficult, the amount and number of specimens limited and these data need to be seen together with the other data presented.

It is unclear or not addressed whether the increase in toxic sRNA is a direct effect of Abeta and tau aggregation in AD or an indirect consequence. The effect of increase in toxic sRNA and Abeta toxicity on cell viability appears to be cumulative rather than interdependent. Alternatively, the mechanism by which the increase of toxic sRNA, including fragmented rRNA or tRNA, is achieved can be discussed in more detail in context of AD to improve the significance of the results obtained in this study. Further, it can also be discussed what kind of cell death pathways are triggered through DISE and how.

We agree that our data cannot exclude at this point a direct or an indirect effect of Abeta and tau aggregation on the toxicity of RISC bound sRNAs. We also agree that the effects could involve both and yes, they could be cumulative. We have clarified this and have added more discussion on the indirect effects on R-sRNAs and the potential role of tRNA and rRNA fragments. We also added a sentence that states that DISE is a combination of multiple cell death pathways and what pathways will be engaged depends on the transcriptome of affected cells. To study each individual cell death pathway ever described in AD and how much may be caused by DISE would be very interesting to address and will be subject of a future study. We have added a sentence to clarify this and added some more detail on what cell death pathways have been reported to contribute to AD.

Where is the data to support the blockage of DNA damage in Abeta treated cells upon enoxacin treatment?

That A β 42 induces DNA damage in differentiated SH cells is well established by comet assays, staining for nuclear foci and through H2AX phosphorylation^{1,2}. We are therefore using γ H2AX Western blotting as a surrogate marker for DNA damage induced by A β 42. Since Enoxacin reduces γ H2AX and DNA damage as already shown by others for cells treated with ionizing radiation (IR)², we concluded that Enoxacin also has that activity in A β 42 induced DNA damage. Our data shown in Fig. 6e clearly demonstrate that treatment with Enoxacin reduces A β 42 induced DNA γ H2AX. However, the experiment did not allow us to distinguish between an effect of Enoxacin on the amount of protective miRNAs in the cells effectively blocking DNA damage or reducing DNA damage through its activity to accelerate DNA repair. The experiment originally shown was done by pretreating the cells with Enoxacin and then adding A β 42 after 24 hours for another 24 hours. We have now added new experiments. We treated differentiated SH cells with A β 42 for 24 hours, resulting in DNA damage, washed out (or not) the A β 42 and then added Enoxacin for another 2 hours. This clearly reduced the level of γ H2AX (new Fig. 6g) suggesting an effect of Enoxacin on accelerating DNA repair, similar to what was reported for cells treated with IR. This experiment does not exclude that Enoxacin also affects DNA damage by toxic sRNAs through increasing nontoxic miRNAs because in contrast to the published experiments that used IR to induced DNA damage, we cannot exclude that A β 42 was not completely removed from the cells.

Minor points

Labelling of blue vs red in seed viability graphs is not always clear (Fig1, Fig4c,d).

We have made it consistent.

Language and punctuation can be improved for clarity.

We have improved that.

Reviewer #2 (Remarks to the Author):

This seems a really interesting pathway, and overall the authors have performed a large amount of work. But an overall issue, which they clearly admit, is that the effect is modest and most of their findings are correlative. So in

the end, the reader is simply left with considering that this pathway appears not very important or impactful for a-beta42 toxicity in Alzheimer's disease.

It is always difficult to establish causality with mouse and human patient brain tissues. This has plagued the AD field for decades and to this day even the assignment of A β plaques as the major initiator of the disease is contested. That is why we added quite a bit of data on the direct response of cells to treatment with toxic A β Os and we feel that these data are highly suggestive of causality.

We do agree that the question of whether the pathway is important for the neurotoxicity of A β 42 or not cannot easily be assessed at this point and needs further study. In our previous work on ovarian cancer³ we have found that even minor imbalances of seed viability of RISC bound sRNAs can have an effect on treatment outcome over a time scale of months. It is therefore conceivable that such an imbalance can have even greater effects in a disease such as AD that affects patients not for months but for many years. In addition, our data strongly suggest that a functional RISC and most likely toxic sRNAs are critical for the DNA damage induced by A β 42. The question then becomes how important DNA damage is in neurodegeneration seen in AD and during aging. An increasing body of literature suggests that DNA damage plays an important role^{4,5}. We have added this to the discussion.

The experiments of Figures 1-4 are correlative, and the effects in any case are modest, so the data are not very compelling that if indeed this pathway is contributing, it is not playing much of a role.

We do not know how much of a shift in seed viability of RISC bound sRNAs is required to negatively impact neurons in a disease that affects patients for many years. Even a minor shift in seed viability over years may make a substantial contribution to the AD etiology just like aging itself. Our data on a drop in seed viability in aged brains are reproducible and we feel are quite compelling.

They try to provide evidence in Figure 5 that specific miRNAs are the most important non-toxic ones in the cells, but one wonders if any miRNA would have the same effect, and the data would be more compelling if they defined target mRNAs that are critical (do the targets function in the pathways of death predicted). In figure 6, the effects of Ago 2 are more considerable, but effects of ATA are rather modest (already previously shown on primary neurons). The most compelling data they present is on DNA damage, but this is rather a side point.

The point of Figure 5 was exactly as requested by the reviewer, not to highlight specific miRNAs but to look at the most abundant nontoxic miRNA species regardless of their established function. The ones we selected to reintroduce into cells were indeed the top three most abundant ones. All three carry nontoxic seeds and their abundance dramatically changed in Drosha ko cells. These three miRNAs alone comprise 21.2% of all miRNAs in the wt cells and only 7.5% and 9.7% in the two Drosha ko clones, respectively. We have added this info to the revised manuscript.

We agree with the reviewers prediction that likely any abundant nontoxic miRNA could play this role. We do not believe that these miRNAs function through targeting (although this will likely also occur) but by outcompeting any sRNA that contains a toxic seed.

Because the activity of ATA is not that well defined, we added the results from Ago2 ko cells. These data are important and clear and the ATA data support this activity.

We do not view the role of toxic RNAs and the RISC in DNA damage as a side point. Growing evidence points at a role of DNA damage in AD and aging^{4,5,6}.

Also knockdown of drosha and ago2 may have broader effects to impact many genes, so a specific effect on DISE is hard to claim.

We agree, that is why all data have to be looked at in combination. Increased sensitivity to A β 42 induce toxicity in Drosha ko cells, reduce sensitivity in ATA treated cells, in Ago2 k.o. cells, and cells with transfected nontoxic miRNAs, together with our other published data strongly points at an involvement of DISE. Because DISE is a network based mechanism, rescue experiments with individual genes is not possible.

Other comments:

For all the graphs, the individual data points should be shown. It is also unclear in a figure like Figure 1e, how many independent experiments do those data represent?

We have added individual data points to all figures where applicable. In addition, we have indicated the number of technical and biological replicates.

Figure 2: The rationale for the rubicon mutant in mice in figure 2 is not clear. They cannot see differences without it, and it should only be impacting autophagy so it is confusing.

Impairing autophagy in this model reduces the ability of cells to clear Abeta resulting in higher amounts of A β O_s ⁷. We have added this information to the revised manuscript. In addition, published work on 5XFAD and 5XFAD Rubicon ko mice shows the difference in measurable cell death in affected brain regions ^{6, 8}. Our data are consistent with DISE contributing to neurotoxicity and a model with little detectable cell death may not involve DISE but also does not accurately model the human disease.

Figure 5: One wonders about including a control miRNA that is not different or changed.

So a random real miRNA. Would addition of any expressed miRNA have the same effect?

The reviewer is correct. Any miRNA can have an effect but others may not be that relevant as they were neither that highly expressed nor changed. The three we used were not selected but the three most abundant miRNAs in the wt cells (see discussion above). In addition, a randomly selected miRNA may actually have an effect on cells through targeting specific mRNAs independent of its seed viability. We therefore decided to use a generic control. We actually first used a control sRNA which also carried a nontoxic seed and was likely incorporated into the RISC. It had some protective effect (data can be provided). Once we realized that, we designed an sRNA with a nontoxic seed that could no longer be loaded into the RISC through chemical modification (used in Fig. 5e) and this finally did not have any activity of reducing the toxicity by A β 42 supporting the model that RISC loading of any sRNA with a nontoxic seed may prevent update ones with toxic seeds shifting the balance to lower seed viability.

Figure 6: at least how it is presented, the different with gamma H2AX is very striking in this figure. That seems a strong effect.

We agree that this could be one of the roles the RISC plays in the mediating the toxicity of A β 42. In addition, we have expanded the analysis of this effect by adding new data.

Reviewer #3 (Remarks to the Author):

1) While Figure 2C provides a diagram of how hexamer ratios might affect cell viability, I think the manuscript would benefit from a diagram in Figure 1 demonstrating how hexameric seeds in miRs act on DICER to alter its function. They could consider simply improving figure 2C and moving it to Figure 1.

We respectfully like to point out that Fig. 2c does not contain a diagram. In case the reviewer is referring to the diagram in Fig. 1c, this figure serves to illustrate our model that it is the ratio of RISC bound sRNAs with nontoxic versus toxic 6mer seeds that determine cell fate. We are not aware of a study that showed that changes in Dicer levels during aging or AD are caused by miRNAs. While this could be a way that Dicer levels are regulated, we do not have any data to support this. In addition, as mentioned in the discussion section miRNA levels could also be regulated by Drosha whose activity has been reported to be regulated by age related stress ^{9, 10}.

2) Fig 2: When discussing the effects of Rubicon KO on 5xFAD, the authors should show the resulting tau pathology and microgliosis in the manuscript – perhaps as supplemental figures.

The mouse brains from tau tg mice were obtained from Dr. Adriana Ferreira. She published that these mice as early as 3 months of age show a substantial amount of cell death in the hippocampus (see Fig. 2C in ¹¹). We agree and we would have wanted to show such data in our manuscript. Unfortunately, this mouse colony was lost during the pandemic and Dr. Ferreira has been trying to rederive these transgenic mice since. Only recently this has worked from cryopreserved embryos but according to her it will take months before the colony has been reestablished. We believe that the published documentation of cell death in this model and information obtained from Dr. Ferreira are sufficient to document that this tauopathy model is characterized by cell death.

3) Fig. 4: The authors should also determine whether DICER transitions to the insoluble fraction with aging, particularly in the AD models. They can also examine human publicly available proteomics data, such as that from Seyfreid's Nat Med paper to examine human data.

It has been reported that in aging mice (2 years versus 3 month old) Dicer expression at the mRNA level is reduced about 90% (see Fig. S2D in ¹²). Transition of Dicer to the insoluble fraction (if it occurs) may therefore only be one mechanism of how miRNA levels are changing during ageing. Also there is convincing data that Drosha can be regulated during aging/AD by being excluded from the nucleus ¹⁰. A change in Dicer or Drosha protein levels has not been reported in the proteomics studies. However, it could be the activity of the proteins or their intracellular localization that is affected and that would not have been detected in these proteomic studies. In addition, there will

likely be multiple mechanisms that regulate miRNA expression levels. One such mechanism is m6A modification of miRNAs (see Ref 77) and we actively studying this.

As pointed out in the Seyfried papers, unbiased molecular profiling data have been obtained at the genomic, transcriptomic, proteomic and metabolomic levels to further understanding of AD pathogenesis. Our work now adds another layer, the role of RISC bound short RNAs. We have added some discussion on the subject and have added citations for two of the Seyfried papers.

4) Fig. 6: I agree that the data show the DNA damage is a result of Abeta tox rather than mediating it. However, the authors should point out that use of enoxacin at 20 uM likely yields extensive off-target activity which complicates interpretation.

We agree and have added this. However, we would like to point out that while drugs such as Enoxacin often have off-target effects, the two different activities of Enoxacin (increase miRNA levels and accelerating DNA damage repair) seem to involve the same target, the Dicer/TRPB complex which is regulating both processes.

5) Do the hexameric repeats produced by C9orf72 affect DISE?

Indeed an interesting question. However, while we have published on Huntington's disease ^{13, 14, 15}, we have not tested whether C9orf72 can produce toxic sRNAs. However, we would like to point out that C9orf72 contains the hexameric repeat GGGGCC and that any sRNA that contains such sequences assuming that part of them can enter the RISC would be expected to be highly toxic to cells as such G-rich sequences produced some of the most toxic seeds (see 6merdb.org). We would like to thank this reviewer for this question and have added this info to the discussion section.

References

1. Li Y, Lu J, Hou Y, Huang S, Pei G. Alzheimer's Amyloid-beta Accelerates Human Neuronal Cell Senescence Which Could Be Rescued by Sirtuin-1 and Aspirin. *Frontiers in cellular neuroscience* **16**, 906270 (2022).
2. Colas J, *et al.* Neuroprotection against Amyloid-beta-Induced DNA Double-Strand Breaks Is Mediated by Multiple Retinoic Acid-Dependent Pathways. *Neural Plast* **2020**, 9369815 (2020).
3. Patel M, *et al.* The ratio of toxic-to-nontoxic microRNAs predicts platinum sensitivity in ovarian cancer. *Cancer Res* **81**, 3985-4000 (2021).
4. Wang H, Lautrup S, Caponio D, Zhang J, Fang EF. DNA Damage-Induced Neurodegeneration in Accelerated Ageing and Alzheimer's Disease. *Int J Mol Sci* **22**, 6748 (2021).
5. Miller MB, *et al.* Somatic genomic changes in single Alzheimer's disease neurons. *Nature* **604**, 714-722 (2022).
6. Soto-Palma C, Niedernhofer LJ, Faulk CD, Dong X. Epigenetics, DNA damage, and aging. *J Clin Invest* **132**, (2022).
7. Heckmann BL, *et al.* LC3-Associated Endocytosis Facilitates beta-Amyloid Clearance and Mitigates Neurodegeneration in Murine Alzheimer's Disease. *Cell* **178**, 536-551 e514 (2019).
8. Oakley H, *et al.* Intraneuronal beta-amyloid aggregates, neurodegeneration, and neuron loss in transgenic mice with five familial Alzheimer's disease mutations: potential factors in amyloid plaque formation. *J Neurosci* **26**, 10129-10140 (2006).
9. Yang Q, *et al.* Stress induces p38 MAPK-mediated phosphorylation and inhibition of Drosha-dependent cell survival. *Mol Cell* **57**, 721-734 (2015).
10. Xu H, *et al.* p38 MAPK-mediated loss of nuclear RNase III enzyme Drosha underlies amyloid beta-induced neuronal stress in Alzheimer's disease. *Aging Cell* **20**, e13434 (2021).

11. Lang AE, Riederer M, Methner DN, Ferreira A. Neuronal degeneration, synaptic defects, and behavioral abnormalities in tau(4)(5)(-)(2)(3)(0) transgenic mice. *Neuroscience* **275**, 322-339 (2014).
12. Mori MA, *et al.* Role of microRNA processing in adipose tissue in stress defense and longevity. *Cell Metab* **16**, 336-347 (2012).
13. Murmann AE, *et al.* Small interfering RNAs based on huntingtin trinucleotide repeats are highly toxic to cancer cells. *EMBO Rep* **19**, e45336 (2018).
14. Murmann AE, Patel M, Jeong SY, Bartom ET, Morton AJ, Peter ME. The length of uninterrupted CAG repeats in stem regions of repeat disease associated hairpins determines the amount of short CAG oligonucleotides that are toxic to cells through RNA interference. *Cell Death Dis* **13**, 1078 (2022).
15. Murmann AE, Yu J, Opal P, Peter ME. Trinucleotide repeat expansion diseases, RNAi and cancer. *Trends in Cancer* **4**, 684-700 (2018).

REVIEWERS' COMMENTS

Reviewer #1 (Remarks to the Author):

The authors have reasonably answered the major and minor points made in the first report. Clarity regarding some of the points has been improved. Specifically, the data regarding the reduction of Abeta-associated DNA damage by enoxacin treatment upon has been supported with new data and is more convincing, even though enoxacin treatment does not impact seed viability. The discussion has also been improved to contextualize the findings of the paper. The authors also acknowledge that the causality between the accumulation of RISC-bound toxic sRNA and Abeta-induced cell death is not easy to establish, with mechanistic efforts outside the scope of this study.

Despite these improvements the overall impact of manuscript is limited to demonstrating the involvement of RISC-bound toxic sRNA in models of Abeta toxicity. This paper would be better suited for a more specialized journal.

Reviewer #2 (Remarks to the Author):

The authors have clearly performed a lot of work, but it seems the manuscript remains largely correlative, and in a number of areas confusing. The first 4 figures remain correlative and the gene knockouts could be affecting many targets. Unfortunately, it is not clear the manuscript has been significantly improved and most of their comments to the reviewers concerns seem to agree that those are concerns. The pathway remains interesting, but perhaps the clarify of manuscript could be improved to provide a more compelling argument.

Reviewer #3 (Remarks to the Author):

This is a resubmitted manuscript. The manuscript addresses a very innovative and intriguing idea, which is that RISC bound short RNAs (sRNAs) that carry G-rich 6mer seeds can kill cells essentially by interfering with other cellular functions. I find this area of research fascinating and am highly interested. However, the experimental approaches used in the manuscript

remain insufficient to clarify the effects of G-rich sRNA toxicity because many of the approaches are correlative or likely to produce an immense number of other effects (e.g., removing DICER, which eliminates miRs) or produce relatively small effects of unclear significance. In addition, the authors appear to insist on calling “Super-Agers” those >80 yrs, when the average age of individuals in the USA is 79 yrs. A super ager is > 95 yrs. The authors refer to “average agers” as “super agers” in the abstract, the introduction, the results and the discussion – despite claiming in the rebuttal that they have addressed this. Also, the authors compare only 3 “Super-agers” (i.e., average agers) to 3 younger donors to draw their conclusions; this type of small cohort is unacceptable for such conclusions. Thus, although I love the idea of this subject area, I find the actual experiments disappointing.